# The *Mycobacterium tuberculosis* methyltransferase Rv2067c manipulates host epigenetic programming to promote its own survival

Prakruti R. Singh [1,2,6], Venkatareddy Dadireddy [3,6], Shubha Udupa[1], Shashwath Malli Kalladi[1], Somnath Shee[4], Sanjeev Khosla[5], Raju S. Rajmani[4], Amit Singh [4], Suryanarayanarao Ramakumar [3] & Valakunja Nagaraja [1,2] ✉

*Mycobacterium tuberculosis* has evolved several mechanisms to counter host defense arsenals for its proliferation. Here we report that *M. tuberculosis* employs a multi-pronged approach to modify host epigenetic machinery for its survival. It secretes methyltransferase (MTase) Rv2067c into macrophages, trimethylating histone H3K79 in a non-nucleosomal context. Rv2067c down-regulates host MTase DOT1L, decreasing DOT1L-mediated nucleosomally added H3K79me3 mark on pro-inflammatory response genes. Consequent inhibition of caspase-8-dependent apoptosis and enhancement of RIPK3-mediated necrosis results in increased pathogenesis. In parallel, Rv2067c enhances the expression of SESTRIN3, NLRC3, and TMTC1, enabling the pathogen to overcome host inflammatory and oxidative responses. We provide the structural basis for differential methylation of H3K79 by Rv2067c and DOT1L. The structures of Rv2067c and DOT1L explain how their action on H3K79 is spatially and temporally separated, enabling Rv2067c to effectively intercept the host epigenetic circuit and downstream signaling.

Pathogen infection leads to the activation of several host defense mechanisms such as immune response, oxidative burst, inflammation, and cell death/survival to curtail the infection[1–3]. These host-pathogen interactions result in the alteration of the transcriptomic landscape of the infected cell leading to selective activation and repression of genes influencing different signaling pathways[4]. Among many, a key mechanism emerging to regulate host transcriptome is the crosstalk between epigenetic modifications and the transcription machinery[1,5]. Epigenetic modifications such as DNA methylation and post-translational modifications (PTM) of histones and nonhistone

chromosomal proteins (NHCP) exert their effect in controlling chromatin dynamics and DNA accessibility to the transcription machinery[6,7]. Histone methylation constitutes one of the crucial controllers of chromatin remodeling. Histone methyltransferases (HMTases) methylate primarily either lysine or arginine, impacting chromatin structure and gene expression. Most of these enzymes catalyze methylation after histones are incorporated into the nucleosomes. For example, Disruptor of Telomeric silencing 1-Like (DOT1L) is a non-SET histone methyltransferase that catalyzes mono-, di- and trimethylation of nucleosomal H3 lysine 79 (H3K79)[8–11]. H2B

---

[1]Department of Microbiology & Cell Biology, Indian Institute of Science (IISc), Bengaluru, India. [2]Jawaharlal Nehru Centre for Advanced Scientific Research (JNCASR), Bengaluru, India. [3]Department of Physics, Indian Institute of Science, Bengaluru, India. [4]Centre for Infectious Disease Research (CIDR), Department of Microbiology and Cell Biology, Indian Institute of Science (IISc), Bengaluru, India. [5]Council of Scientific and Industrial Research-Institute of Microbial Technology, Chandigarh (CSIR -IMTech), Chandigarh, India. [6]These authors contributed equally: Prakruti R. Singh, Venkatareddy Dadireddy. ✉e-mail: vraj@iisc.ac.in

ubiquitination stimulates DOT1L activity , and the methylation of H3K79 leads to a variety of effects that include DNA damage response, humoral immune response, CD4[+] and CD8[+] T cell differentiation, embryonic development, cell cycle progression, and somatic reprogramming[12–14]. Such dynamic changes in histone methylation and other epigenetic marks are also seen in response to external stimuli, including bacterial infection. Manipulation of host epigenome by the pathogen upon infection ensuing alteration in cellular pathways to gain an advantage is an emerging phenomenon [2,15–17].

*Mycobacterium tuberculosis* (*Mtb*), a successful intracellular pathogen and a leading cause of mortality[18], has evolved diverse mechanisms in response to multiple stresses encountered upon infection. It inhibits oxidative and nitrosative stress, suppresses apoptosis and immune response, and manipulates the host ubiquitin system for its enhanced survival[19–23]. About 5% of the *Mtb* genome encodes epigenetic modifiers that include kinases, methyltransferases, acetyltransferases, and succinyltransferases, hinting at their role in the manipulation of epigenetic signaling post-infection (p.i). A few studies indicate the role of these modifiers in fine-tuning the mycobacterial epigenome[24–28]. However, a majority of them remain uncharacterized and their involvement in modifying bacterial or host epigenome is yet to be understood.

Here, we have investigated the structure and function of a *Mtb* methyltransferase (MTase) Rv2067c. Our study revealed that Rv2067c is secreted by *Mtb* into the macrophages, where it exerts its function by at least two mechanisms. First, it trimethylates free histone H3 at lysine 79 (H3K79), ultimately triggering the changes in the expression of a gene subset in the host. Second, it represses host MTase DOT1L expression pre-empting DOT1L activity on H3K79 in nucleosomal context. We provide the structural basis for how Rv2067c methylates free histone H3 and not nucleosomal H3 by comparing its structure with that of the nucleosome-bound DOT1L, which catalyzes H3K79 methylation only in the nucleosomal context. This multi-pronged epigenetic strategy elicited by *Mtb* through its effector leads to a simultaneous increase in necrosis and inhibition of apoptosis, ensuring the pathogen's survival.

## Results

### Rv2067c methylates Histone H3

Rv2067c was identified as one of the interacting partners of *Mtb* histone-like protein HU (MtHU) in pulldown experiments with MtHU as a bait (Supplementary Fig. 1a). Sequence analysis of Rv2067c revealed the presence of S-Adenosyl-L-methionine (SAM) dependent MTase domain suggesting Rv2067c is an MTase (Supplementary Fig. 1b). However, on performing MTase assays, Rv2067c did not methylate recombinant MtHU (Supplementary Fig. 1c). As MtHU carboxy-terminal domain (CTD) shows remarkable similarity with histone tails[29], we investigated whether mammalian histones are targets for the MTase activity of Rv2067c. In MTase assays using salt-extracted histones from THP-1 monocytes as substrates, Rv2067c as MTase, and tritiated SAM as a methyl donor, methylation of histones was detected (Fig. 1a and Supplementary Fig. 1d). To determine which of the histones is a substrate for Rv2067c, MTase assays with individual histones were performed. Among the four histones, only H3 was methylated by Rv2067c (Fig. 1b, c). Using tandem MS/MS, the site of methylation was identified as lysine 79 of H3 (H3K79). A mass shift of 42 Da at lysine 79 of H3 implied Rv2067c trimethylates H3K79 (Fig. 1d). To ascertain, MTase assays were carried out with unmodified 73–83 amino acid (aa) peptide of H3, followed by dot blot. H3K79me3 antibody detected a signal, confirming Rv2067c trimethylates H3K79 (Fig. 1e, Supplementary Data 1). Further, when K79 of H3 was mutated to alanine, H3A79 protein was not methylated by Rv2067c (Fig. 1f and Supplementary Fig. 1e), re-confirming its specificity for lysine 79 and no other H3 lysine residues as a substrate for the MTase. To demonstrate the SAM-dependent activity of Rv2067c, the conserved GxG motif was mutated

to RxR, and MTase assays showed that the mutation abolished the activity of Rv2067cRxR (Fig. 1g and Supplementary Fig. 1f).

Intriguingly, H3K79 methylation is a conserved epigenetic mark in eukaryotes catalyzed by DOT1L. Yeast Dot1 and hDOT1L catalyze mono-, di- and trimethylation of H3K79, but only in the nucleosome core particle (NCP), and free H3 is not the substrate[8,30,31]. To determine whether Rv2067c can also methylate nucleosomal H3, we performed MTase reactions with H3 alone, reconstituted H3-H4 dimer, histone octamers, and NCPs (Supplementary Fig. 1g, h). Rv2067c methylated only free H3 and did not methylate H3 in octamer or nucleosomal core (Fig. 1h, Supplementary Fig. 1i).

### Rv2067c trimethylates H3K79 upon *Mtb* infection

Having established H3 as a substrate of Rv2067c, we investigated the interaction and activity of the MTase with endogenous H3 by transfecting HEK293T with pcDNA: Rv2067c-SFB (S-protein, FLAG, streptavidin-binding peptide). Immunoblots showed H3 interacts with Rv2067c (Fig. 2a). Interaction between Rv2067c and H3 was reconfirmed by performing immunoprecipitation with H3 antibody (Supplementary Fig. 2a, Supplementary Data 1). Enhanced H3K79me3 mark in cells expressing Rv2067c compared to Rv2067cRxR ("Methods", SAM-binding mutant) indicated methylation by the MTase (Fig. 2b, c). To probe Rv2067c function upon *Mtb* infection, knockout (ΔRv2067c), complemented (ΔRv2067c:comp), overexpression (Rv2067c:OE) and ΔRv2067c:RxR ("Methods") strains were constructed. The schematic depicts knockout generation strategy (Fig. 2d). The strains were confirmed by PCR (Supplementary Fig. 2b–d, Supplementary Data 2), qRT-PCR (Supplementary Fig. 2e), and immunoblotting with Rv2067c antibody (Fig. 2e and Supplementary Fig. 2f). These strains did not show any growth difference in vitro (Supplementary Fig. 2g). When macrophages were infected with these strains, enhanced H3K79me3 mark was observed for wildtype *Mtb* (Wt*Mtb*) and ΔRv2067c:comp, with no change in the knockout (Fig. 2f) and ΔRv2067c:RxR strain (Fig. 2g), 24 h post-infection (h.p.i).

Although the above data showed enhanced H3K79me3 upon infection, the subcellular context of methylation was not clear. After its synthesis in the cytoplasm, H3 enters the nucleus either in free form[32] or as H3-H4 heterodimers[33,34] before assembling into octamers and NCP. To investigate the cellular location of methylation of H3 by Rv2067c, nuclear and cytoplasmic extracts were analyzed for H3K79me3 mark in THP-1 post-infection (p.i). Enhanced H3K79me3 was observed in the nuclear extract for Wt*Mtb* and ΔRv2067c:comp infected macrophages. No signal for H3K79me3 was observed in the cytosolic fraction, which could be due to the rapid nuclear import of newly synthesized histones and incorporation into chromatin[32] (Fig. 2h). To capture H3 methylation in the cytosol, if any, cells were co-transfected with FLAG-tagged H3 (H3-FL) and Rv2067c. In both nuclear and cytoplasmic extracts, H3K79me3 was observed, indicating H3 methylation in both compartments (Fig. 2i). Thus, it is apparent that Rv2067c, a SAM-dependent MTase, not only trimethylates H3K79 in vitro but also upon transfection and *Mtb* infection in macrophages.

### Rv2067c is secreted into the host macrophages

Since Rv2067c trimethylates H3K79 in the host, we examined its secretion from mycobacteria. Rv2067c is secreted into the culture filtrate of Wt*Mtb* and THP-1 macrophages p.i (Supplementary Fig. 3a, b). As *M. smegmatis* (*M.smeg*) does not have a homolog for the MTase, its secretory nature was investigated by ectopically expressing Rv2067c (*M.smeg*:Rv2067c-FLAG). It was detected in the culture filtrate by immunoblotting using Rv2067c antibody and mass spectrometry (Supplementary Fig. 3c–e). Confocal microscopy p.i revealed MTase's localization both in the cytoplasm and nucleus of macrophages infected with Rv2067c:OE and *M.smeg*:Rv2067c-FLAG (Supplementary Fig. 3f, g). Subcellular fractionation of HEK293T transfected with Rv2067c confirmed its localization in cytoplasm and nucleus

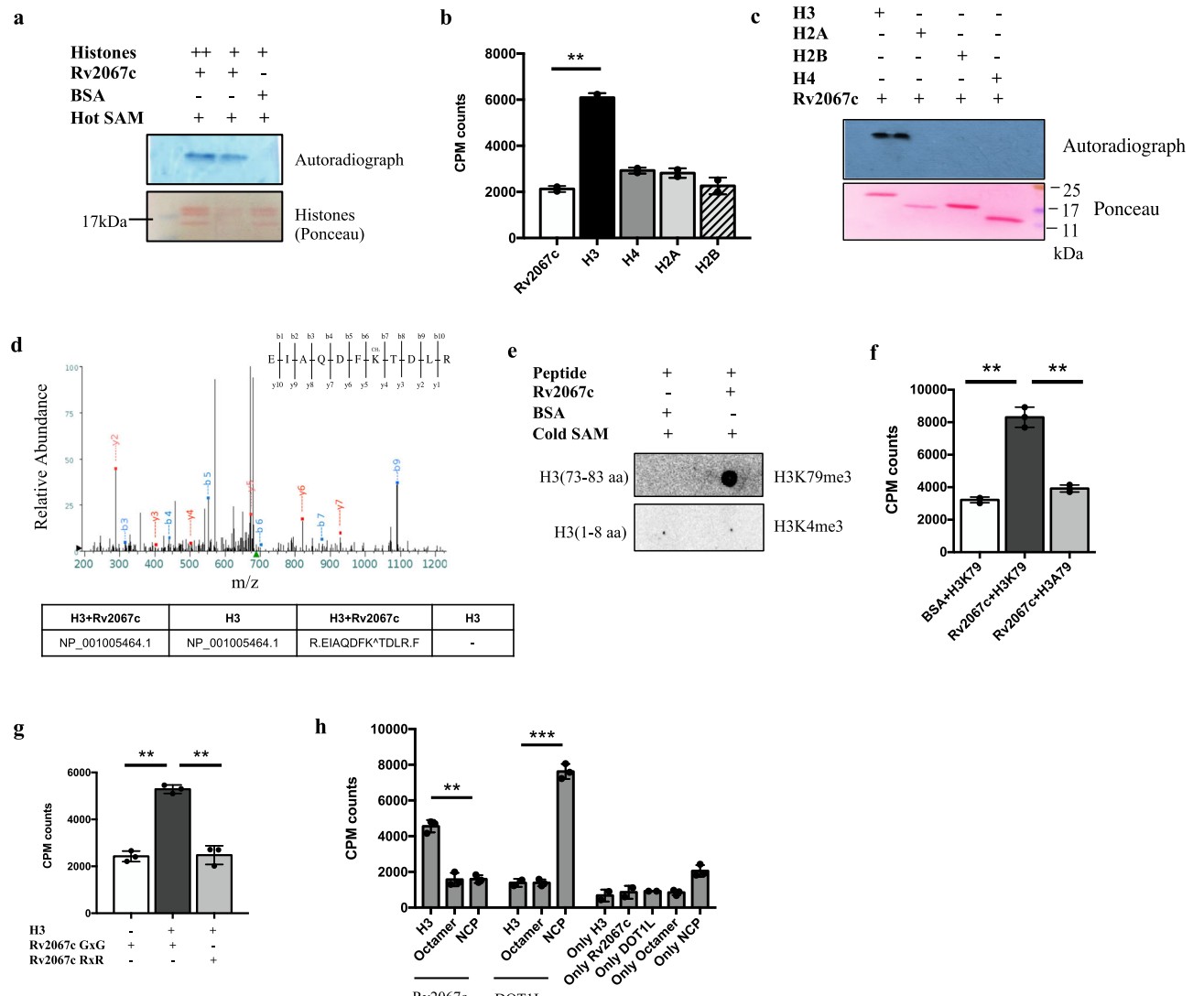

**Fig. 1 | Rv2067c in vitro methylates histone H3 at lysine 79. a** Autoradiograph of methylation by Rv2067c with two different concentrations of histones salt extracted from THP-1 monocytes as substrate and tritiated S-Adenosyl-L-methionine (³H-SAM) as a methyl group donor (Lane 1 and 2). BSA (lane 3) was kept as negative control. Lower panel shows ponceau staining ($n = 2$ independent experiments). **b, c** Bar graph and autoradiograph showing methyltransferase (MTase) activity of Rv2067c against recombinant H3, H2A, H2B and H4 as substrate and tritiated SAM as methyl group donor. Recombinant Rv2067c was kept as negative control for scintillation counts and ponceau staining shows the loading of histones for autoradiograph. ($n = 2$ independent experiments; $p = 0.002$). **d** MS/MS spectra of H3K79 trimethylation performed by Rv2067c on recombinant H3. The table shows peptide EIAQDFK^TDLR is trimethylated (^ represents trimethylation) when the assay was performed with H3 and Rv2067c. H3 alone serves as control ($n = 1$). **e** Blots probed with H3K79me3 and H3K4me3 antibodies respectively depict

MTase assays with synthetic H3 peptide 73–83 aa and 1–8 aa (negative control) as substrates, Rv2067c as MTase and non-radioactive SAM as methyl donor. **f** Scintillation counts for Rv2067c MTase activity with recombinant H3 and H3A79 mutant as substrate ($n = 3$ independent experiments; $p = 0.002, 0.003$). **g** Graph shows MTase assays with Rv2067c and its SAM-binding mutant (GxG to RxR; "Methods") using recombinant H3 as substrate ($n = 3$ independent experiments; $p = 0.006, 0.008$). **h** Comparison of MTase activity of Rv2067c and DOT1L. Rv2067c methylates free H3, whereas DOT1L methylates H3 in nucleosomal core particle (NCP). Recombinant H3, reconstituted octamers and NCPs were kept as controls; Rv2067c and DOT1L were kept as control to check auto-methylation ($p = 0.006, 0.0005$). The data for MTase assays is plotted as mean. Error bars correspond to SD and *P*-value was calculated using unpaired two-tailed Student's *t*-test. **, *P*-value < 0.01, ***, *P*-value < 0.001.

(Supplementary Fig. 3h). The predicted Nuclear Localization Signal (NLS) was located towards the C- terminus of Rv2067c. The deletion of carboxy-terminal 30 aa resulted in the retention of MTase in the cytoplasm, as revealed by immunostaining and subcellular fractioning (Supplementary Methods and Supplementary Fig. 3i, j).

### Structure of Rv2067c differs from that of DOT1L
To understand the structural differences conferring the context-dependent methylation of H3K79 by DOT1L and Rv2067c, and the basis of free H3 methylation by Rv2067c, we determined the

structure of Rv2067c to 2.40 Å resolution (Supplementary Note 1, and Supplementary Table 1). Each monomer (407 aa) of Rv2067c homodimer (Fig. 3a, Supplementary Fig. 4a, Supplementary Methods, Source Data and Supplementary Software File 1) is composed of three domains: N-terminal SAM-binding catalytic domain (CD), dimerization domain (DD), and C-terminal domain (CTD) (Fig. 3b–d). The catalytic domain contains a central seven-stranded β-sheet flanked by α-helices, a characteristic of seven-β-strand (7BS) MTases (class-I MTases)[35] (Fig. 3c and Supplementary Fig. 4b, c). The DD is composed of two subdomains separated on the

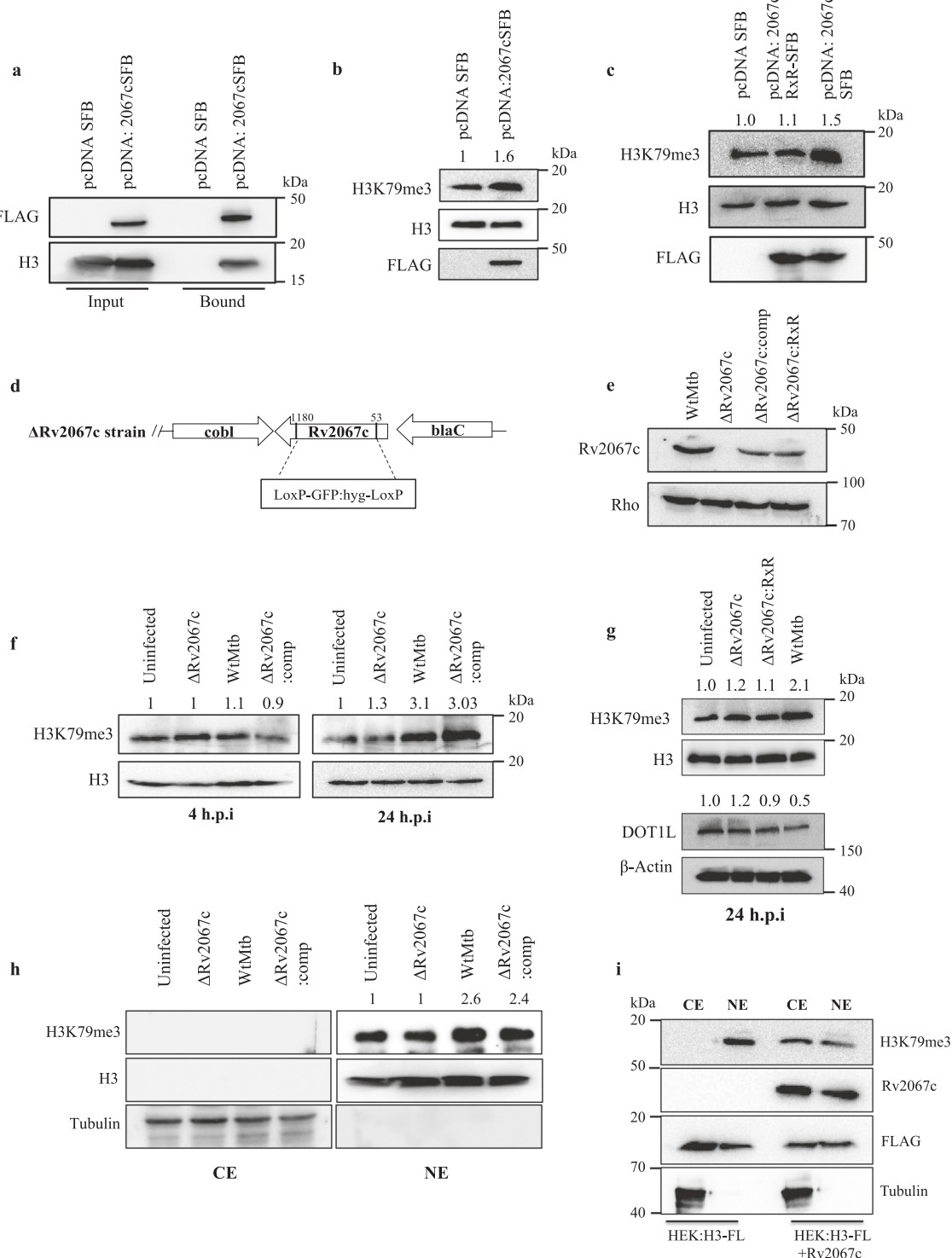

primary structure, namely the large subdomain (LSD; residues 163-230, four α-helices) and the small subdomain (SSD; residues 257-291, two α-helices), which are inserted into the catalytic domain at β5-α5 and β6-β7 loops, respectively (Fig. 3c, d and Supplementary Fig. 4b). Dimer is formed by the cross-subdomain interactions between the DDs of the two monomers A and B in a fashion: $LSD_A$-$SSD_B$ and $LSD_B$-$SSD_A$, and $LSD_A$-$LSD_B$ (Fig. 3b and Supplementary Fig. 4c). CTD is about 100 residues long with α + β architecture (Fig. 3c, d and Supplementary Fig. 4b, c). The dimerization interface, in conjunction with the other two domains, forms two parallel trough-like structures across the dimer, which constitute the acceptor substrate-binding trough (Figs. 3a, b and 4).

In contrast to dimeric Rv2067c, DOT1L is a 1537 aa long monomer with catalytically active region spanning the first 416 residues ($DOT1L^{cat}$), of which residues 4–332 are structured ($DOT1L^{3D}$). $DOT1L^{3D}$ is an elongated structure having an N-terminal portion and a SAM-binding catalytic core with a 7BS MTase fold (Fig. 3d–e and Supplementary Fig. 4b, c). $DOT1L^{3D}$ is followed by a positively charged lysine-rich region (residues 386–416) involved in DNA binding and a disordered region with interspersed coiled-coil domains that aid in protein-protein interactions[8,11,36,37] (Fig. 3d, f). Thus, barring a common class-I MTase fold (Supplementary Fig. 4c), Rv2067c and DOT1L markedly differ in their primary structure, domain composition, architecture, and oligomeric state.

**Fig. 2 | Rv2067c methylates histone H3 at lysine 79 upon infection in THP-1 macrophages. a** Western blot showing interaction between Rv2067c and H3. Pulldown with streptavidin beads on lysates of HEK293T transfected with pcDNA: Rv2067cSFB and pcDNA SFB (control). 5% of whole cell lysate was kept as Input. Blot was probed with FLAG and H3 antibodies. SFB stands for S-protein, FLAG, streptavidin-binding peptide. **b, c** H3K79me3 mark in cell lysates of HEK293T transfected with pcDNA: Rv2067cSFB and pcDNA SFB (**b**); pcDNA: Rv2067cRxR-SFB (lane 2) (**c**). Blots were probed with H3K79me3, stripped, and probed with H3 antibody used as loading control. FLAG indicates expression of Rv2067c. **d** Schematic of the strategy for the generation of from *Rv2067c* gene knockout. From +53 bp to +1180 bp of was deleted Rv2067c and replaced with hygromycin resistance cassette (LoxP-GFP:hyg-LoxP). **e** Immunoblot shows levels of Rv2067c in the lysates of Wt*Mtb* (lane 1), ΔRv2067c (lane 2), ΔRv2067c:comp (lane 3) and ΔRv2067c:RxR (lane 4). Blot was probed with Rv2067c and Rho (control) antibodies. **f** H3K79me3 mark in THP-1 cell lysates 4 and 24 h.p.i with various *Mtb* strains. Lane 1: uninfected THP-1 macrophages; Lane 2, 3 and 4: THP-1 macrophages infected with ΔRv2067c, Wt*Mtb* and ΔRv2067c:comp strains respectively for each time point. H3 was used as loading control. **g** H3K79me3 mark (upper panel) and DOT1L levels (lower panel) in THP-1 infected with ΔRv2067c:RxR (lane 3). H3 and β-Actin were used as loading control, respectively. **h** H3K79me3 mark in subcellular fractions of uninfected THP-1 and macrophages infected with ΔRv2067c, Wt*Mtb* And ΔRv2067c:comp strains respectively, 24 h.p.i. Tubulin and H3 were used as control for cytosolic and nuclear extracts, respectively. NE nuclear extract, CE cytosolic extract. **i** Immunoblots show detection of H3K79me3 mark in cytosolic and nuclear extracts of HEK293T expressing Rv2067c and FLAG-tagged H3(H3-FL). HEK293T cells expressing HEK:H3-FL were kept as control. Blots were probed with FLAG and Rv2067c antibodies to show expression of H3 and Rv2067c, respectively. Tubulin was used as control for cytosolic fraction. **e–h** Comp stands for complemented. Values above the blot represent quantitation (arbitrary units). For immunoblots, *n* = 2 independent experiments. Source data are provided as a Source Data file.

## Rv2067c structure precludes nucleosomal H3K79 methylation

Apart from the insights obtained from the gross structural differences, the mechanism of nucleosomal methylation by DOT1L and the comparison of active sites of DOT1L and Rv2067c provided the basis for substrate-context-dependent methylation by Rv2067c. During methylation, DOT1L binds across the nucleosome (Nuc) or ubiquitinated nucleosome (UbNuc) and interacts with different regions of the Nuc or UbNuc. These include (1) hydrophobic interaction between the C-terminal helix of DOT1L[3D] and ubiquitin, (2) contact between DOT1L R282 and H2A-H2B acidic patch, (3) tucking of basic residues of H4 tail into the acidic groove of DOT1L N-terminal portion, and (4) interaction of DOT1L F131 and W305 loops with H3K79 loop (Supplementary Fig. 5). These structural appendages of DOT1L that aid in nucleosomal methylation and their equivalences in Rv2067c are different with respect to the primary sequence and tertiary structure (Supplementary Fig. 6). In addition, DOT1L binding stimulates a conformational transition in H3K79 loop which places H3K79 in the active site of DOT1L from its inaccessible conformation, which is pivotal for methylation[9–11,38] (Supplementary Fig. 5).

Besides the differences in the structural elements of DOT1L and Rv2067c, the active site architectures of Rv2067c and DOT1L are starkly different. The active site of DOT1L is a narrow tunnel, formed by three loops viz. a loop connecting the N-terminal portion and catalytic core (F131 loop), β4-α4 and β6-β7 (W305 loop), enough to accommodate H3K79 side chain. At one end of the tunnel lies the methyl moiety of SAM, and the other end is open for H3K79 entry[8,11] (Fig. 4a and Supplementary Fig. 7). In contrast, Rv2067c contains a substrate-binding trough formed by all three domains (Figs. 3b and 4b). The active site lies at one end of the trough, deep inside the protein, encompassed by CD and LSD (Fig. 4b). Such an active site architecture presumably obstructs the accessibility by a bulky substrate like nucleosome but could allow extended substrates like free H3. By a computational procedure (rotation scan) using a hypothetical enzyme-substrate reaction-complex model (Supplementary Methods), the accessibility of the Rv2067c active site by nucleosome was assessed with a supposition that the active-state H3K79 conformation exists in nucleosomes. The rotation scan (Supplementary Methods) calculated a minimal number of 211 atoms involved in atomic clashes in the Rv2067c-nucleosome reaction complex and in the corresponding pose, nucleosome approaches Rv2067c active site via an unconventional direction, the SAM entry site instead of substrate-binding trough (Fig. 4c). In short, due to the lack of nucleosome interacting structural appendages and different active site architecture, Rv2067c is incompetent to methylate nucleosomal H3K79, but its substrate-binding trough could accomodate free H3 for methylation.

## Active site dynamics of Rv2067c facilitate free H3 methylation

The substrate-binding trough of Rv2067c resembles the substrate-binding cleft of rickettsial protein lysine methyltransferase 1 and 2 (PKMT1/2). These two MTases methylate multiple lysines of the outer membrane protein B (OmpB) [39–41] (Supplementary Note 2). Inspection of the Rv2067c substrate-binding trough revealed that the residues facing the methyl moiety of SAM (SAM-facing residues) occlude the H3K79 binding (Fig. 4b). Sequence analysis showed that the SAM-facing residues are part of a conserved patch that encompasses SAM (Supplementary Fig. 8a–c, Supplementary Note 3 and Supplementary Methods). Structural analysis showed that the active site in protein MTases lies nearly perpendicular to the SAM-binding pocket and accommodate the substrate residue (Supplementary Fig. 9a–h), which in the case of Rv2067c is occluded. The relatively high-temperature factors (Supplementary Fig. 10, Supplementary Data 3 and Supplementary Software File 2) of highly conserved Y20 and less conserved Q228 residue indicate the dynamic nature of the active site. The active site dynamics were tested by performing all-atom molecular dynamics (MD) simulations. The substrate-binding trough of Rv2067c in the simulation trajectory was analyzed using POVME 3.0 (Supplementary Methods). During 100 ns long simulations, widening of the occluded active site was observed that makes room for H3K79 (Fig. 5a–c and Supplementary Movie 1). In addition to the active site dynamics, the dα3-dα4 loop (residues 209–214, part of LSD and the substrate-binding trough) near the active site is disordered, as evident from the missing electron density in the crystal structure and relatively high root mean square fluctuations observed from simulations (Fig. 3a, b, Supplementary Fig. 11, Source Data and Supplementary Software File 3). Thus, the dynamic active site and the flexible dα3-dα4 loop are likely to facilitate H3 binding and methylation.

Based on the nature of the substrate-binding trough, active site dynamics, and mode of substrate binding to the active site, we modeled the Rv2067-H3 complex (Supplementary Methods). Due to the lack of tertiary structural information of monomeric H3 (Supplementary Note 4), we chose an 11-mer H3 peptide 73-EIAQDFKTDLR-83 which is a competent substrate for Rv2067c (Fig. 1e). This peptide region assumes a random coil structure in free H3 (Supplementary Fig. 12a, b)[42,43]. Due to the directional ambiguity, as the peptide binds in its extended conformation in a narrow trough, we propose two modes of peptide binding, i.e., from N- to C-terminal or from C- to N-terminal direction with respect to the active site (Fig. 6 and Supplementary Data 4). The binding conformations resemble the binding of histone tails to SET domain MTases[44,45] or protein arginine N-MTases (PRMTs)[46]. In both modes, the side chain of H3K79 occupies the active site and aligns nearly perpendicular to the SAM axis (Fig. 6), as seen in protein MTase complexes (Supplementary Fig. 9), amenable for methylation.

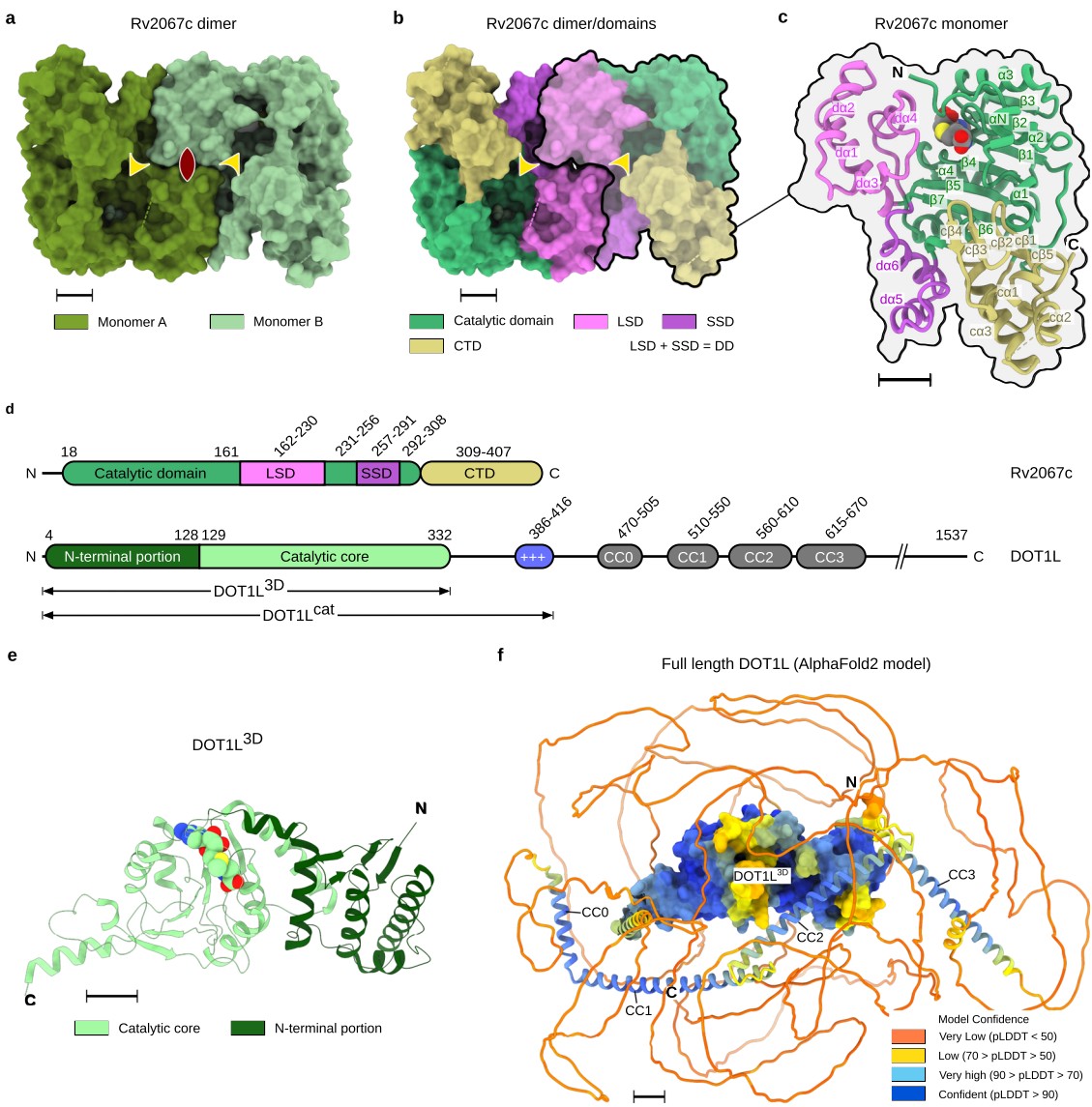

**Fig. 3 | Structure of Rv2067c and its comparison to DOT1L. a** Crystal structure of Rv2067c. The asymmetric unit contains two molecules of Rv2067c related by two-fold symmetry and the two-fold axis is perpendicular to the page, shown as biconvex sign (maroon). **b, c** Each monomer is composed of catalytic domain (CD), dimerization domain (DD, composed of large subdomain, LSD and small subdomain, SSD) and C-terminal domain (CTD). Dimer is formed by cross-subdomain interactions between the DDs. SAH is depicted as spheres. **a, b** The broken lines represent the missing electron density (disordered regions). The substrate-binding troughs are indicated with arrowheads (yellow). **d** Schematic of domain organization in Rv2067c (top) and DOT1L (bottom). **e** Crystal structure of DOT1L (PDB: 1NW3). Residues 5–332 are structured (DOT1L³ᴰ). The DOT1L³ᴰ contains an N-terminal portion and a catalytic core. SAM is depicted as spheres. **f** Full length human DOT1L (1537 aa) structure model (AlphaFold2 model, AF-Q8TEK3-F1). The structured part of the catalytic domain (DOT1L³ᴰ) is rendered as surface and the remaining region as cartoon, which is disordered with interspersed α-helices (CC0–CC3) that form coiled-coil (CC) structures with the interacting partners. The model is colored according to the AlphaFold2 prediction confidence. The low confidence score indicates intrinsic disorder. The N- and C-termini are labeled as N and C, respectively. SAH S-adenosyl-L-homocysteine, SAM S-adenosyl-L-methionine. Scale bars 10 Å.

## Rv2067c modulates expression of DOT1L

The increase in H3K79me3 upon infection is likely due to the activity of Rv2067c. Alternatively, the increase could be DOT1L specific as both MTases target the same site in H3. Hence, we examined the expression of DOT1L upon *Mtb* infection. Surprisingly, DOT1L expression was reduced in macrophages infected with Wt*Mtb* and Rv2067c:comp, 12 and 24 h.p.i (Fig. 7a, b and Supplementary Fig. 13a) and also in HEK293T transiently expressing Rv2067c (Supplementary Fig. 13b). However, DOT1L expression did not alter in macrophages infected with ΔRv2067c (Fig. 7a, b) and ΔRv2067c:RxR (Fig. 2g; lower panel). These results imply that the higher level of H3K79me3 is Rv2067 specific.

To ascertain that the enhanced H3K79me3 mark is due to Rv2067c MTase activity, we inhibited DOT1L in two ways - RNA interference and a small molecule inhibitor. HEK293T transfected with DOT1L shRNA showed reduced DOT1L expression (Supplementary Fig. 13c, d); a corresponding reduction of H3K79me3 mark was confirmed by immunoblotting (Fig. 7c, center lane). Cells co-transfected with DOT1L shRNA and pcDNA:Rv2067c showed an increase in H3K79me3 mark, confirming the gain of mark is due to the MTase activity of Rv2067c (Fig. 7c). In the second approach, DOT1L was inhibited using EPZ004777[47], a chemical derivative of SAM which binds to DOT1L with high affinity (Kd = 0.1 ± 0.02 nM) to competitively inhibit SAM binding[48]. EPZ004777 did not affect Rv2067c activity even at higher concentrations (50 nM) (Fig. 7d). Since EPZ004777 specifically inhibits DOT1L and not Rv2067c, macrophages were treated with EPZ004777 prior to infection. While the H3K79me3 levels were reduced in uninfected inhibitor-treated cells, the methylation levels

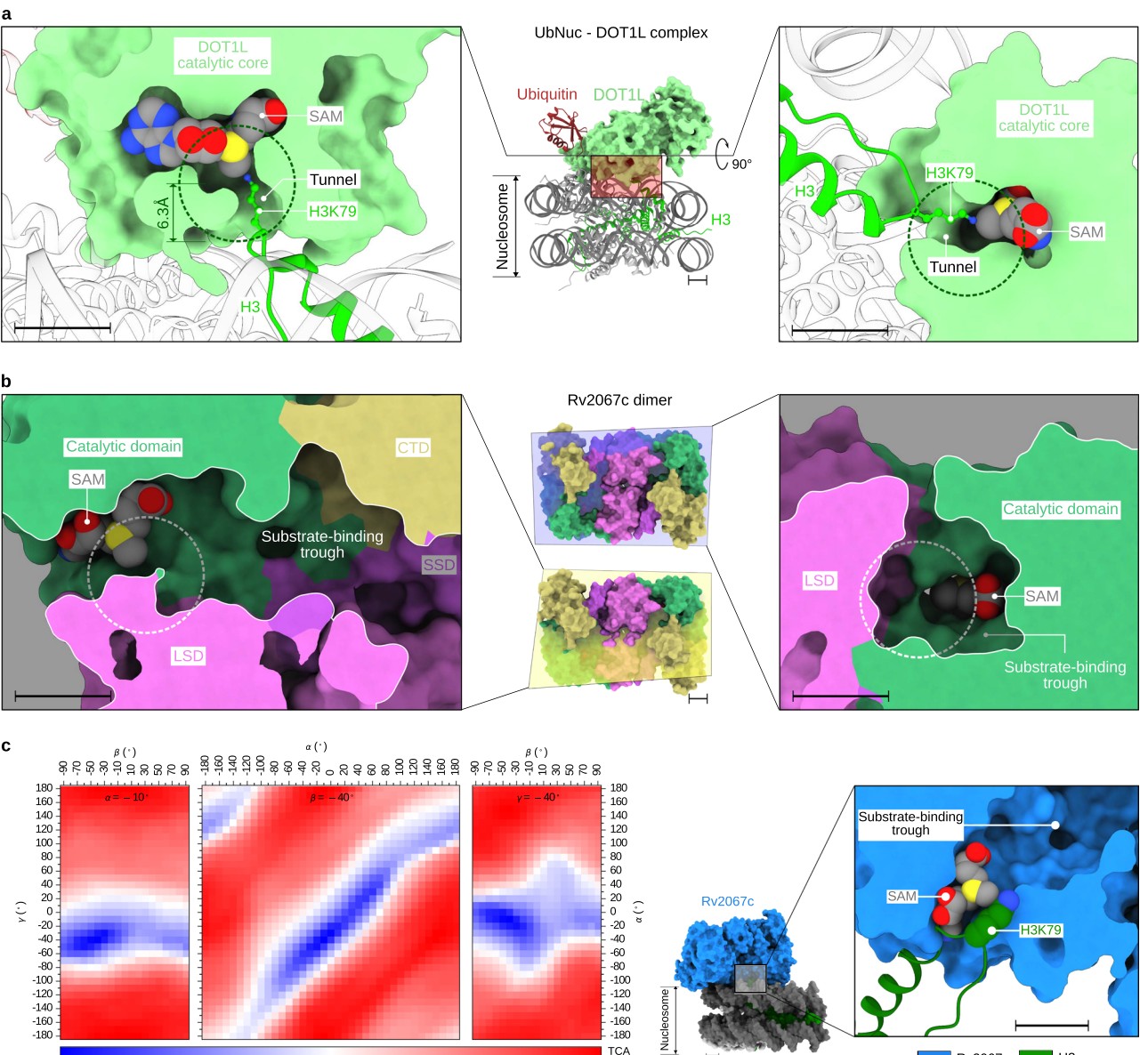

**Fig. 4 | Comparison of active sites between Rv2067c and DOT1L. a** The complex between ubiquitinated nucleosome (UbNuc) and DOT1L$^{cat}$ (middle panel). The substrate lysine (nucleosomal H3K79) binds in a narrow tunnel (cross sections across SAM, left and right panels). The length of the tunnel is about the length of the lysine side chain in its extended rotamer. **b** Cross sections (left and right panels) of substrate-binding trough of Rv2067c as depicted by planes passing through Rv2067c dimer (middle panel). The active site (broken circle) lies deep inside (right panel) and at one end (left panel) of the trough, and encompassed by catalytic domain and LSD. The active site, region opposite to methyl group of SAM is occluded and leaves no room for substrate lysine binding. **a, b** The radius 6.3 Å of the circle is equal to the length of lysine side chain in its extended rotamer. The substrate lysine must reach the encircled region to undergo methylation. **c** Non-

accessibility of Rv2067c active site by nucleosomal H3K79 is shown by a hypothetical reaction complex model of nucleosome-Rv2067c and rotation scan. The total number of clashing atoms (TCA) for each pose is shown as a function of elemental rotation angles (α, β and γ) (left panel). Each subplot corresponds to the value of either α or β or γ at minimum TCA value (i.e., 211). The pose between Rv2067c and nucleosome reaction complex with minimum TCA (right panel). In this pose, nucleosomal H3K79 accesses the reaction center of Rv2067c via the SAM entry site. SAM: S-adenosyl-L-methionine, LSD: Large subdomain of dimerization domain, SSD: Small subdomain of dimerization domain, CTD: C-terminal domain. Scale bars 10 Å. The Source Data and plotting script for TCA plot are available at https://doi.org/10.5281/zenodo.8352903 and https://github.com/Venkat-Dadi/Rotation_Scan.

were restored in Rv2067c infected macrophages, despite the reduction of DOT1L RNA, 24 h.p.i (Fig. 7e and Supplementary Fig. 13e). Together, these two approaches confirm that the increased H3K79me3 is specifically due to Rv2067c.

Next, to understand the basis of DOT1L repression upon *Mtb* infection, we analyzed epigenetic marks on the DOT1L gene. Enhanced H3K9me3, H3K27me3, and H3K79me3 and depleted H3K4me3 marks were observed at the two loci in the gene- peak 1(chr19:2164620-2168335) and peak 2(chr19:2169309-2181752)[49,50] in macrophages infected with Wt*Mtb* and ΔRv2067c:comp but not with ΔRv2067c

(Supplementary Fig. 13f–j). The enhanced H3K79 mark in DOT1L inhibitor-treated cells upon infection further confirms that the K79 mark at the loci is due to Rv2067c. Moreover, the increase in the repressive H3K9me3 and H3K27me3 mark on both loci upon *Mtb* infection indicates that Rv2067c brings about reduced expression of DOT1L.

**H3K79me3 mark by Rv2067c alters host gene expression**
To identify the genes impacted by H3K79 methylation by Rv2067c, we performed ChIP with H3K79me3 antibody on DOT1L knockdown HEK

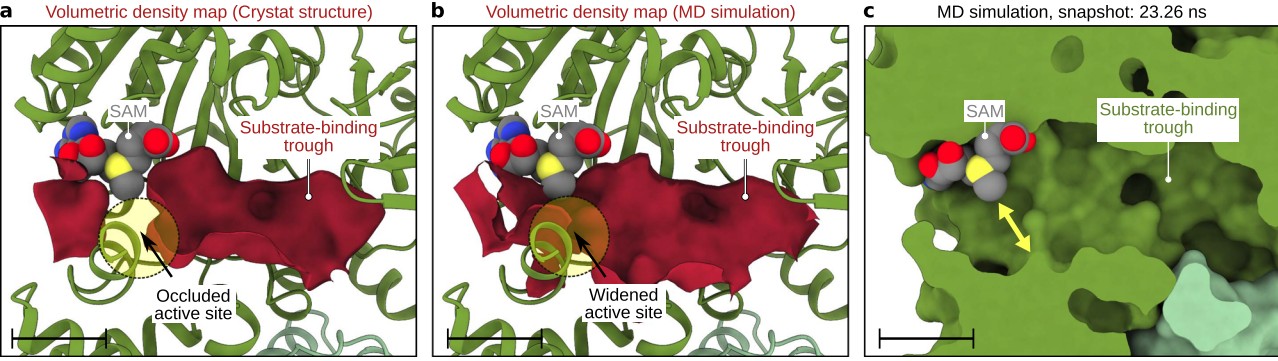

**Fig. 5 | Rv2067c active site dynamics.** Volumetric density map of Rv2067c substrate-binding trough calculated using POVME 3.0 for a crystal structure (monomer A) and molecular dynamics (MD) trajectory. **a** In the crystal structure, the active site is occluded (broken circle). **b** During 100 ns MD simulations, the active site was widened to accommodate substrate lysine (broken circle). The

volumetric density map is shown as an isosurface (maroon) at 0.1 contour level. **c** The widened active site (marked with a double-headed yellow arrow) as compared to the occluded active site seen in the crystal structure (Fig. 4b) for a simulation snapshot is shown. SAM S-adenosyl-L-methionine. Scale bars 10 Å.

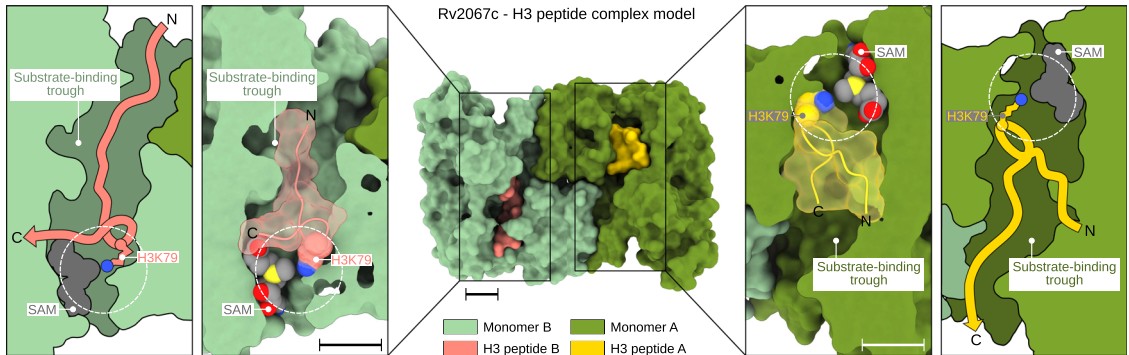

**Fig. 6 | Rv2067c-H3 peptide complex model.** Rv2067c-H3 peptide complex model (middle panel) depicting H3 peptide (73-EIAQDFKTDLR-83) binding in the substrate-binding trough. The H3 peptide was modeled in two binding modes along the trough, i.e., from N- to C-terminal (left panel) or from C- to N- terminal (right panel) with respect to the active site (broken white circle). In both modes,

H3K79 can access the SAM methyl group in a similar fashion seen in MTase-substrate complexes (Supplementary Fig. 9a–h). SAM S-adenosyl-L-methionine. Scale bars 10 Å. The coordinates for the Rv2067c-H3 peptide model are available as Supplementary Data 4.

cells expressing Rv2067c (HEK-DOT1L-KD; see "Methods"). DOT1L knockdown was confirmed by western analysis (Supplementary Fig. 14a). Among several identified, five loci where the H3K79 mark was originally absent in CD14 monocytes[49] were chosen for analysis (Supplementary Table 2). The appearance of H3K79me3 on *TMTC1, SESTRIN3*, and *NLRC3* loci suggests the methylation is Rv2067c specific (Supplementary Fig. 14b). This enrichment of H3K79 mark was seen even when DOT1L was not inhibited in cells transfected with Rv2067c (Supplementary Fig. 14b). The increased expression of these genes in cells expressing Rv2067c indicates that H3K79me3 added by Rv2067c is an activating mark (Supplementary Fig. 14c). Similarly, macrophages infected with Wt*Mtb* and the MTase complemented knockout strain (ΔRv2067c:comp) showed increased H3K79me3 mark and enhanced expression of TMTC1, SESTRIN3, and NLRC3 (Fig. 8a, b) along with a concomitant change in SERCA2 expression (Supplementary Fig. 14d). In contrast, infection with ΔRv2067c and ΔRv2067c:RxR showed decreased H3K79me3 and reduced expression of these three genes (Fig. 8a, b; Supplementary Fig. 14e, f).

### DOT1L repression impacts cytokine response and cell death pathways

Reduced DOT1L expression in macrophages infected with *Mtb* expressing Rv2067c indicated another role for Rv2067c in manipulating host epigenetic circuitry. Recruitment of DOT1L to the

promoter of specific cytokines results in the addition of the H3K79me2/3 mark facilitating their expression[51]. Hence, reduced DOT1L due to Rv2067c action could lead to the downregulation of DOT1 responsive cytokines. Transcriptomic analysis of THP-1 macrophages infected with *M.smeg*:Rv2067c-FLAG and *M.smeg* (Supplementary Fig. 14g, h) showed 3497 and 1444 genes were significantly upregulated and downregulated, respectively (Fold Change ≥ 1.5) (Source Data). Among these innate immunity, cytokine activity, chemokine-related signaling pathway genes were downregulated[52] (enrichment score >5 and *P*-value <0.05) (Supplementary Fig. 14i). Notably, expression of pro-inflammatory cytokines IL-6, TNF-α, subunits of IL12 – IL12A and IL12B along with DOT1L were downregulated (Supplementary Fig. 14j). Importantly, reduced expression of these cytokines is seen upon *Mtb* infection (Wt*Mtb* and ΔRv2067c:comp), 24 h.p.i (Fig. 8c). Further, promoters of IL-6 and TNF-α showed reduction in H3K79me3 mark in macrophages infected with Wt*Mtb* and ΔRv2067c:comp and not with ΔRv2067c (Fig. 8d).

TNF-α is known to induce programmed necrosis as well as caspase-8 mediated apoptosis upon *Mtb* infection[53,54]. We observed increased cleavage of caspase-8 along with decreased RIPK3 in macrophages infected with ΔRv2067c in comparison to Wt*Mtb* and ΔRv2067c:comp (Fig. 8e). RIPK1 and FADD expression did not alter (Fig. 8e). Notably, anti-apoptotic markers BCL-2 and BCL-xL were downregulated in ΔRv2067c but not in Wt*Mtb* and ΔRv2067c:comp

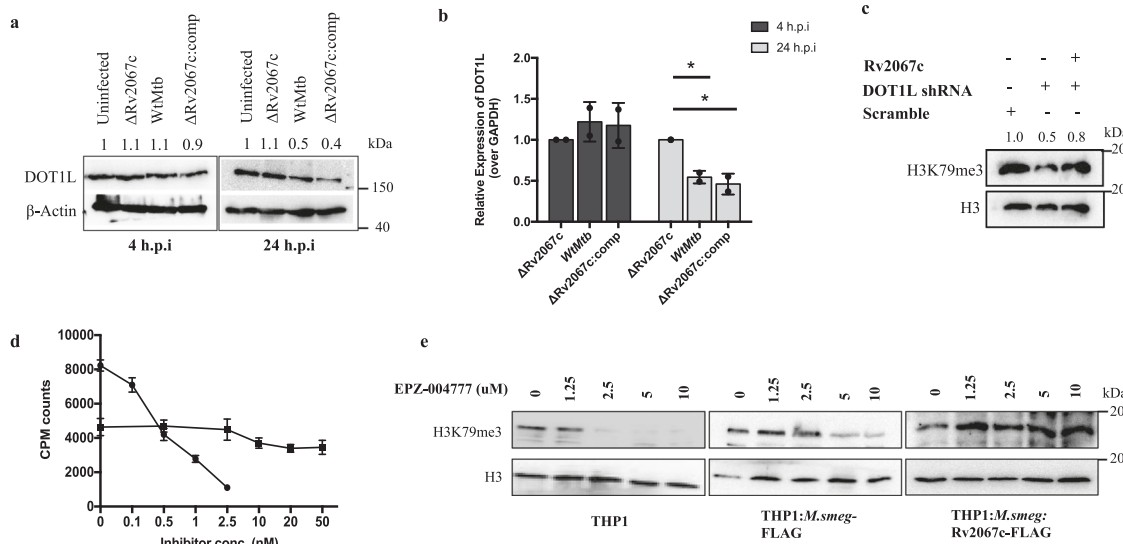

**Fig. 7 | Rv2067c modulates DOT1L expression. a** Immunoblot depicts DOT1L levels in THP-1 cell lysates 4 and 24 h.p.i with various *Mtb* strains. Lane 1: uninfected THP-1; Lane 2, 3 and 4: THP-1 infected with ΔRv2067c, Wt*Mtb* and ΔRv2067c:comp strains, respectively, for each time point. β-Actin was used as loading control. Values above the blot represent quantitation (arbitrary units). **b** Relative expression of DOT1L in THP-1 4 (dark gray bars) and 24 (light gray bars) h.p.i with ΔRv2067c, Wt*Mtb* and ΔRv2067c:comp normalized with respect to uninfected THP-1. Levels were normalized against GAPDH. *n* = 2 independent infections; *p* = 0.02, 0.04. Data is plotted as mean and error bars represent SD. *P*-values depicted on the graphs were calculated using unpaired two-tailed Student's *t*-test; *P*-value < 0.05. Comp stands for complemented **c** H3K79me3 mark in cell lysates of HEK293T with the following treatment: Lane 1: scramble shRNA (control); Lane 2: DOT1L inhibition by

shRNA and Lane 3: DOT1L inhibition by shRNA along with expression of Rv2067c. H3 was used as loading control. Values above the blot represent quantitation (arbitrary units). **d** Graph depicts inhibition of DOT1L with small molecule inhibitor EPZ004777 which does not inhibit Rv2067c. Filled circles and squares represent scintillation counts for DOT1L and Rv2067c with EPZ004777 inhibitor at indicated concentrations. Each data point represents the mean of three replicates at each specified concentration of the compound, and the error bars represent SD. **e** Immunoblot analysis of H3K79me3 mark in uninfected THP-1 (left panel), inhibitor-treated THP-1 infected with *M.smeg*:FLAG (middle panel) and *M.smeg*:Rv2067c-FLAG (last panel) at indicated concentrations. H3 was kept as loading control. Source data are provided as a Source Data file.

infected macrophages. BID (uncleaved) decreased and BAX levels did not alter in macrophages infected with ΔRv2067c (Fig. 8f). Flow cytometry analysis showed a higher percentage of macrophages infected with Wt*Mtb* and ΔRv2067c:comp undergo necrosis (Fig. 8g and Supplementary Fig. 14k), suggesting that Rv2067c restricts host-mediated apoptosis, promoting necrosis.

Next, we investigated the role of Rv2067c in enhancing the survival of intracellular bacilli upon infection. Both ΔRv2067c and ΔRv2067c:RxR had significantly reduced bacterial survival 24 and 48 h.p.i in macrophages compared to Wt*Mtb* and ΔRv2067c:comp (Fig. 9a). Survival advantage was also observed for *M.smeg*:Rv2067c-FLAG in macrophages (Supplementary Fig. 14l). Notably, the bacterial burden of ΔRv2067c reduced in the lungs of BALB/c mice 8 weeks p.i when compared to Wt*Mtb and* ΔRv2067c:comp (Fig. 9b). The gross and histopathological changes in the lungs of the ΔRv2067c infected mice 8 weeks p.i were comparable with the bacillary load observed (Fig. 9c, d). The extent of pulmonary tissue destruction was lowest in the lungs of mice infected with ΔRv2067c (score 4) relative to Wt*Mtb* (score 11) and ΔRv2067c:comp (score 11) (Fig. 9e). Altogether, these results confirm that Rv2067c is a determinant for *Mtb's* intracellular survival.

## Discussion

Intracellular pathogens engage the host with diverse strategies for infection, survival, and multiplication. One emergent strategy is to circumvent and outwit the host response by interfering with the host epigenetic machinery. We have uncovered mechanisms by which a secreted MTase Rv2067c of *Mtb* alters the host epigenome ensuring intracellular survival of the pathogen. Rv2067c adds H3K79me3 mark on H3 and represses host MTase DOT1L upon *Mtb* infection to alter host gene expression. In doing so, it pre-empts DOT1L-catalyzed H3K79me3 in the nucleosomal context. Consequently, DOT1L-

mediated pro-inflammatory cytokines and apoptotic pathways are attenuated. By adding the H3K79me3 mark on loci where the DOT1L-specific mark is absent, Rv2067c activates the expression of genes involved in countering the host response to the pathogen. A combination of these actions by the MTase steers the host signaling events away from apoptosis and toward necrosis.

Intracellular pathogens have adopted successful strategies to perturb host epigenetic pathways providing glimpses into underlying molecular mechanisms[16,17,55,56]. The MTase RomA of *Legionella pneumophila* methylates H3K14 in the octamer or nucleosome to impact the expression of host innate immunity genes[17]. *Mtb* MTase Rv1988 adds an H3R42me2 mark to repress NOS2, NOXA1, NOX1, and NOX2, but the context of methylation (free or nucleosomal) remains to be investigated[55]. Eis, a GNAT family acetyltransferase of *Mtb*[24], facilitates evasion of autophagy by increasing IL-10 expression through acetylation of H3 at its promoter to activate Akt/mTOR/p70S6K pathway [57]. In addition, dysregulated expression of histone deacetylase and sirtuins is seen in macrophages after *Mtb* infection resulting in an altered epigenetic landscape[58–63]. *Mtb* infection also results in the induction of SET8, a SET domain containing MTase, which methylates H4K20me1 to regulate apoptosis and subsequent inflammation assisting *Mtb* survival[64]. Other *Mtb* virulence factors also carry out host epigenetic reprogramming. Secretory protein ESAT-6 remodels the host chromatin and inhibits interferon (IFN)-γ-induced type I as well as type IV CIITA[65].

However, unlike these examples, Rv2067c employs multiple approaches to alter host cellular signaling. By adding H3K79me3 mark on TMTC1, SESTRIN3, and NLRC3, which were devoid of H3K79me3 mark before *Mtb* infection, Rv2067c enhances their expression. Upregulation of SESTRIN3 results in scavenging ROS after primary ROS accumulation, thereby delaying apoptosis following *Mtb* infection[66]. NLRC3 negatively regulates NF-κB and CD4[+] T cell response

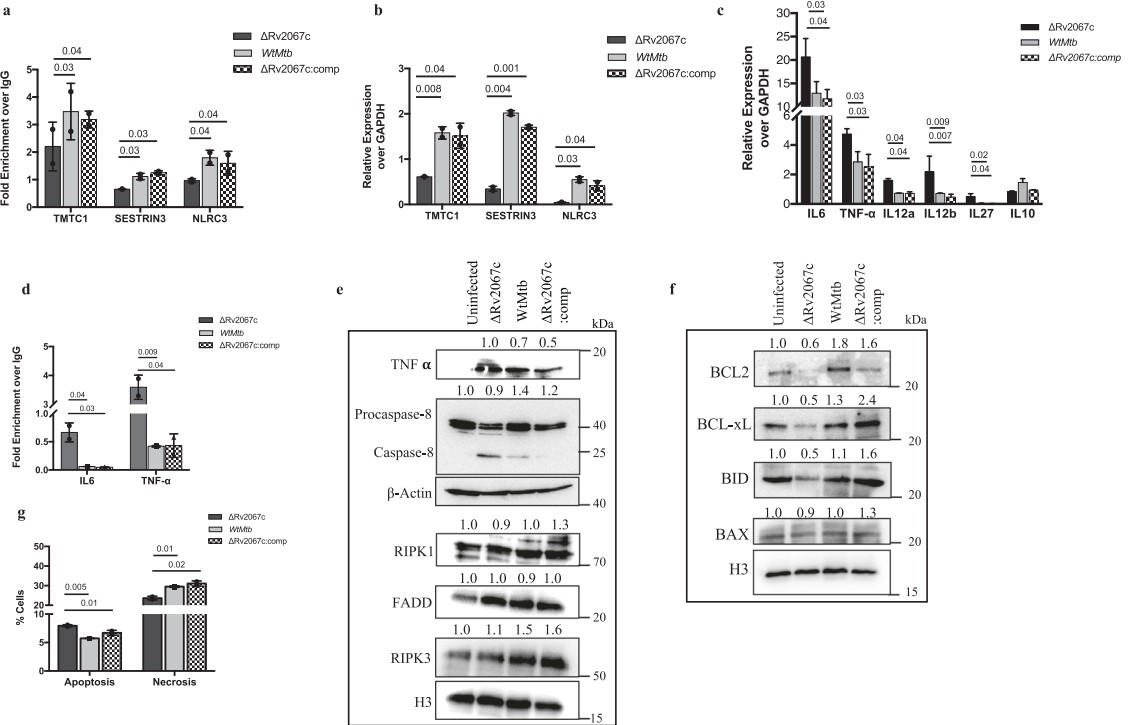

**Fig. 8 | Downstream events consequent to H3K79 methylation by Rv2067c.**
**a** Fold enrichment of H3K79me3 at specific loci (as indicated) added by Rv2067c in THP-1 infected with ΔRv2067c (dark gray), Wt*Mtb* (light gray) and ΔRv2067c:comp (checker). Data is shown as fold enrichment over IgG control. **b** Relative expression of TMTC1, NLRC3 and SESTRIN3 in THP-1 infected with ΔRv2067c (dark gray), Wt*Mtb* (light gray) and ΔRv2067c:comp (checker) with respect to uninfected THP-1 macrophages. Ct values were normalized against GAPDH. **c** Graph shows relative expression of indicated cytokines in THP-1 infected with ΔRv2067c (dark gray), Wt*Mtb* (light gray) and ΔRv2067c:comp (checker). Ct values were normalized against GAPDH. **d** Fold enrichment of H3K79me3 mark at the promoter region of IL-6 and TNF-α in THP-1 infected with ΔRv2067c (dark gray), Wt*Mtb* (light gray) and ΔRv2067c:comp (checker). Data is shown as fold enrichment over IgG control. **a**–**d** For all ChIP and qRT-PCR, *n* = 2 independent infections. Data is plotted as mean

and error bars represent SD. *P*-values depicted on the graphs were calculated using unpaired two-tailed Student's *t*-test. **e**, **f** Immunoblots depict levels of caspase-8, necrotic, pro- and anti-apoptotic markers as indicated in THP-1 cell lysates infected with various *Mtb* strains. Lane 1: uninfected THP-1; Lane 2, 3 and 4: THP-1 infected with ΔRv2067c, Wt*Mtb* and ΔRv2067c:comp strains, respectively. β-Actin was used as loading control for TNF-**α** and caspase-8 and H3 as loading control for RIPK1, FADD and RIPK3 (**e**). H3 was used as loading control (**f**). Values above the blot represent quantitation (arbitrary units). **g** Graph depicts the percentage of apoptotic and necrotic macrophages analyzed by flow cytometry upon infection with ΔRv2067c (dark gray), Wt*Mtb* (light gray) and ΔRv2067c:comp (checker) *n* = 2 independent infections. Data is plotted as mean and error bars represent SD. *P*-values depicted on the graphs were calculated using unpaired two-tailed Student's *t*-test. Source data are provided as a Source Data file.

suppressing innate immunity, thereby promoting *Mtb* survival[67]. A previous study showing activation of NF-κB in macrophages infected with Rv2067c transposon mutant indicated the role of Rv2067c in suppressing NF-κB[68], supporting our findings. TMTC1, an endoplasmic reticulum integral membrane protein, influences calcium homeostasis by interacting with SERCA2 and steers the cell toward a necrotic pathway[53]. Thus, it is apparent that the H3K79me3 mark added by Rv2067c enhances the expression of genes involved in overcoming host response to infection. In parallel, by repressing DOT1L and the consequent reduction in DOT1L-specific H3K79me3 mark, Rv2067c plays a role in reduced expression of the IL-6 and TNF-α. Recruitment of DOT1L to the proximal promoter of IL-6 and the addition of the H3K79me2/3 mark facilitates its transcription activation[51]. We show that by regulating TNF-α expression, Rv2067c reduces caspase-8 cleavage along with upregulation of RIPK3 in macrophages infected with *Mtb*. Concomitantly, the expression of anti-apoptotic markers BCL-2 and BCL-xL is enhanced. Following *Mtb* infection, Bcl-xL and RIPK3 are critical for preventing caspase-8-mediated apoptosis and induction of necrosis in the mitochondria of macrophages [69,70]. Thus, inhibition of caspase-8-mediated apoptosis by the MTase and parallel promotion of expression of anti-apoptotic genes and RIPK3 benefits the pathogen's intracellular survival. Infection with pathogens can trigger varied levels of TNF-α expression[71]. Although intracellular TNF-α concentration can determine the outcome of *Mtb* infection [53,72], Rv2067c action involving several downstream signaling events enables

the pathogen to overcome host response. Hence, multiple strategies orchestrated by *Mtb* through Rv2067c appear to be a fail-safe mechanism to ensure its success (Fig. 10).

The context-dependent specificity for H3K79 of DOT1L and Rv2067c stems from the different structural architecture of these two MTases despite the fact that they belong to class-I MTases. The structure of monomeric DOT1L is tailored to recognize H3 only in the nucleosomes, whereas the dimeric Rv2067c contains a confined active site toward the end of the narrow substrate-binding trough that rejects histone dimer/tetramer and nucleosome but can accept extended substrates such as free H3. The Rv2067c-H3 peptide model that is consistent with the mode of substrate binding observed in MTases explains the methylation of free H3 by Rv2067c. Through its interacting partners, DOT1L is targeted to a particular locus in the chromatin where it installs methylation marks[36,73]. Ubiquitin and possibly other interacting partners facilitate H3K79 methylation by DOT1L[11,74]. In contrast, the structure of Rv2067c allows it to methylate newly synthesized H3 in the cytoplasm and possibly in the nucleus. This would affect the fate of chromatin regions where the newly methylated H3 is incorporated in nucleosomes.

Being a regulator of host innate immune defense, DOT1L function on nucleosomal H3 upon pathogen infection would enable immediate activation of pro-inflammatory pathways for bacterial clearance. Thus, despite methylating the same target as DOT1L, *Mtb's* epigenetic writer, by its distinct structure, warrants spatial and temporal separation in its

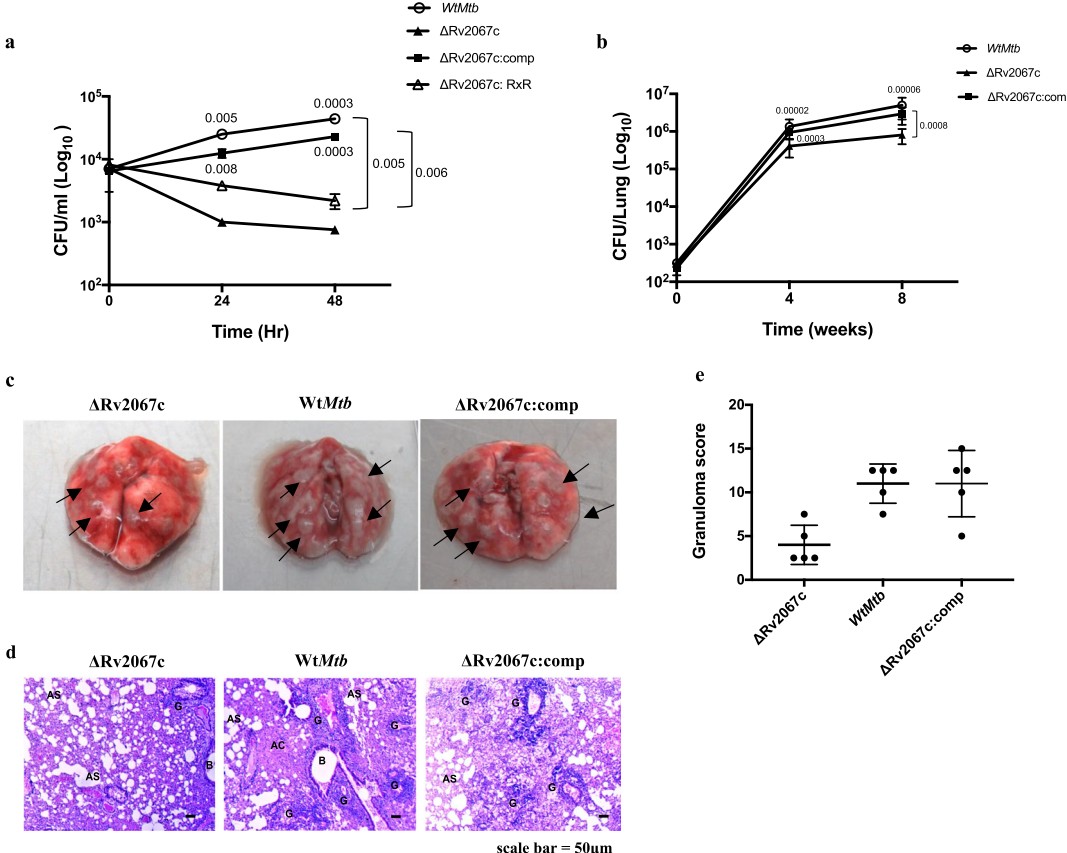

**Fig. 9 | Rv2067c gives survival advantage to *Mtb*. a** Survival of intracellular bacilli post 24 and 48 h of infection in THP-1 with ΔRv2067c (filled triangle), Wt*Mtb* (open circle), ΔRv2067c:RXR (open triangle) and ΔRv2067c:comp (filled square). *n* = 3 independent infections. Three technical replicates were kept during each infection. **b** Bacillary burden in lungs of mice infected with ΔRv2067c (filled triangle), Wt*Mtb* (open circle) and ΔRv2067c:comp (filled square), 4- and 8-week p.i (*n* = 6 animals per group per time point, two technical replicates for plating were kept). **a**, **b** Data is plotted as mean and error bars represent SD. *P*-values depicted on the graphs were calculated using unpaired two-tailed Student's *t*-test. **c** Gross pathology of the lungs of BALB/c mice infected with ΔRv2067c, Wt*Mtb* and ΔRv2067c:comp harvested 8 weeks after infection. **d** Haematoxylin and eosin–stained mouse lung sections 8 weeks p.i with ΔRv2067c, Wt*Mtb* and ΔRv2067c:comp.The pathology sections show granuloma (G), alveolar space (AS), Alveolar consolidation (AC) and bronchial lumen (B). All images were taken at 4X magnification and scale bars = 50 μm. **e** Graph depicts granuloma score of lung sections of mice infected with ΔRv2067c, Wt*Mtb* and ΔRv2067c:comp, 8 weeks p.i. (*n* = 5 animals). The error bars represent SD. Source data are provided as a Source Data file.

action to reprogram cellular response. Although a few host enzymes post-translationally modify newly synthesized H3 and H4[75,76], the methylation of cytosolic H3 by Rv2067c appears to be a clever strategy of *Mtb* while encountering the host. By intercepting nucleosomal methylation of H3K79me3 and pre-emptive cytosolic methylation of H3, it provides the pathogen multiple options to manipulate downstream pathways. As free H3 exists both in cytoplasm and nucleus[32,77], localization of Rv2067c in the nucleus suggests the possibility of either free H3 methylation in the nucleus or yet to be identified other substrates of the MTase.

Finally, DOT1L is a versatile MTase participating in various cellular functions in diverse tissues regulating effective humoral immune response, CD4⁺ and CD8⁺ T cell differentiation, embryonic development, cell cycle progression, somatic reprogramming, and DNA damage [12–14]. Notably, DOT1L also affects virus multiplication by enhancing the antiviral response[78] and limits nematode multiplication by controlling CD4⁺ T cell activation[79]. Inhibition or silencing of DOT1L during viral infection resulted in decreased nuclear translocation of NF-κB [78]. Thus, as a regulator of the innate immune response, DOT1L seems to be an integral component of the host defense arsenal against a variety of infections. Targeting the DOT1L expression and its function, simultaneously methylating the same residue on H3, the MTase of the pathogen seems to act as a dual-edged sword. Although the present study shows the effect of Rv2067c in macrophages, it is likely to impact DOT1L function and host signaling in other cells, which are also targets for *Mtb* infection. Given its multiple roles elicited in promoting *Mtb* survival, Rv2067c is yet another virulence determinant of the pathogen. Thus, Rv2067c could be an attractive target for developing new lead molecules to curtail *Mtb* infection.

## Methods
### Study approval
The research complies with all relevant ethical regulations. All protocols for mice and rabbit experiments described in this study were approved by the Institutional Animal Ethical Committee (IAEC), Indian Institute of Science (IISc), Bangalore, India.

### Growth conditions
*E. coli* cultures were grown in Luria-Bertani (LB) medium (BD Biosciences). *M. smegmatis* (*M.smeg*) was cultured in 7H9 media (BD Biosciences) and *Mtb* strains were cultured in 7H9 media supplemented with 1X OADC, 0.1% Tween-80, and 0.2% glycerol. When required, the culture medium was supplemented with hygromycin (50 μg/ml for mycobacteria, 150 μg/ml for *E. coli*), kanamycin (25 μg/ml for mycobacteria and 50 μg/ml for *E. coli*), and ampicillin (100 μg/ml).

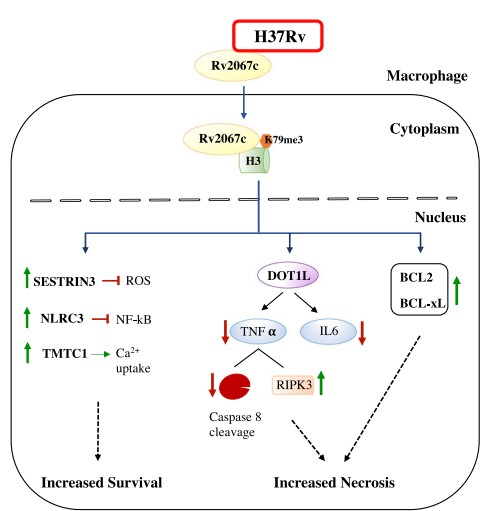
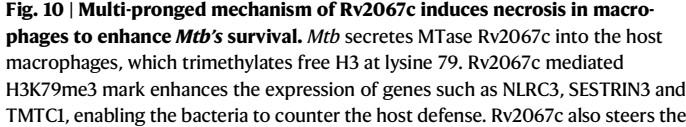
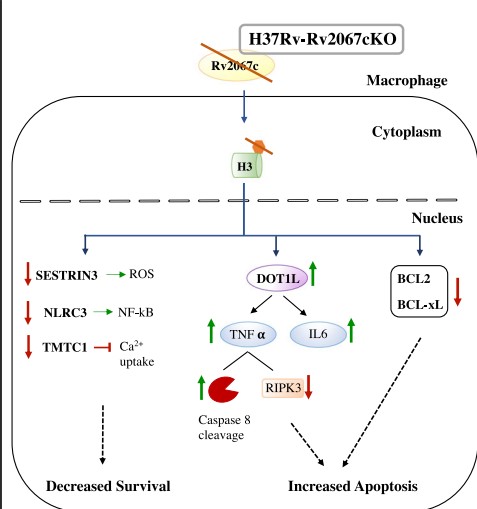

**Fig. 10 | Multi-pronged mechanism of Rv2067c induces necrosis in macrophages to enhance *Mtb's* survival.** *Mtb* secretes MTase Rv2067c into the host macrophages, which trimethylates free H3 at lysine 79. Rv2067c mediated H3K79me3 mark enhances the expression of genes such as NLRC3, SESTRIN3 and TMTC1, enabling the bacteria to counter the host defense. Rv2067c also steers the cell toward RIPK3-mediated necrosis by downregulating DOT1L to impact the expression of pro-inflammatory cytokines such as TNF-α and IL-6. A concomitant upregulation of anti-apoptotic markers is observed. Green and red arrows show upregulation and downregulation of genes, respectively.

## Crystallization of Rv2067c
Crystallization trials were conducted by sitting drop vapor diffusion method at 22 °C, with each drop containing 0.8 µl of tagless protein (20 mg/ml) and 0.8 µl of crystallization solution. Rod-shaped crystals were obtained in less than 24 h, in conditions (from crystallization screen Morpheus, Molecular Dimensions, UK) containing buffer 1 (0.1 M imidazole/MES acid, pH 6.5) or buffer 2 (0.1 M sodium HEPES/MOPS acid, pH 7.5) with precipitant mix (20% w/v polyethylene glycol 500* monomethyl ether + 10% w/v polyethylene glycol 20,000) and additive mix NPS (0.03 M sodium nitrate + 0.03 M sodium phosphate dibasic + 0.03 M ammonium sulfate) or halogens (0.03 M sodium fluoride + 0.03 M sodium bromide + 0.03 M sodium iodide) or carboxylic acids (0.02 M sodium formate + 0.02 M ammonium acetate + 0.02 M sodium oxalate). For experimental phasing, crystals were either grown in the presence of sodium iodide (NaI) or 5-amino-2,4,6-triiodoisophthalic acid (I3C) or soaked in mother liquor containing NaI or I3C.

## Data collection, processing, and structure determination
Diffraction data were collected at MX beamline ID29, European Synchrotron Radiation Facility (ESRF), Grenoble. For each dataset, 3600 images were collected with 0.1° oscillation at an X-ray wavelength of 1.7000 Å (anomalous) or 1.0723 Å (native). All the datasets were collected at cryogenic temperature (100 K). Diffraction images were processed using the XDSAPP3.0[80]. The best anomalous signal was obtained from crystals grown in condition: buffer 1 with precipitant mix and NPS and soaked in a mother liquor containing 150 mM NaI for 5 min. The anomalous dataset was processed to 3.25 Å resolution. A better resolution (2.40 Å) dataset with a weak anomalous signal from crystals grown in the same condition in the presence of 10 mM I3C was processed as a native dataset. No native crystals diffracted better than 2.40 Å. Both datasets belonged to the P4$_1$2$_1$2 space group. Initial phases were obtained by single anomalous dispersion (SAD) method with iodine as a marker atom, using CRANK2[81] from CCP4i2 suite[82]. A near-complete model was built using Buccaneer[83]. The 2.40 Å resolution structure was determined by PHASER's molecular replacement (MR) module[84] using phased structure as the MR search model. Alternate cycles of model building and refinement were carried out using Coot[85] and Refmac5[86], respectively. Model quality was assessed using MolProbity[87].

## Cell culture and macrophage infection
THP-1 was acquired from ATCC (Manassas, VA). THP-1 monocytes were cultured in RPMI medium supplemented with 10% fetal bovine serum at 37 °C and 5% CO$_2$. Monocytes were differentiated for 18 h with 10 ng of PMA, and after 24 h of recovery in PMA-free RPMI medium, infection was performed. Differentiated THP-1 macrophages are referred to as macrophages in the manuscript.

For infection, bacterial strains were grown up to an OD$_{600}$ of 0.6–0.8, washed once with phosphate-buffered saline (PBS) and resuspended in medium. Infections were carried out at 1:10 MOI for *M.smeg* and 1:5 MOI for *Mtb* strains respectively, unless otherwise mentioned. After 4 h of infection, gentamycin (50 µg/µl) was added to kill any extracellular bacteria. Media was changed after 1 h of gentamycin treatment, and cells were incubated for the required time.

Cells were harvested after washing with ice-cold PBS and lysed in 1X RIPA buffer (50 mm Tris-HCl, pH 7.4, 1% Nonidet P-40, 0.25% sodium deoxycholate, 150 mm NaCl, 1 mm EDTA). Cell lysates were spun at 12,000×*g* for 20 min at 4 °C. Supernatant was collected and 50–100 µg was loaded on SDS-PAGE for western analysis. RNA was isolated from cells using TRI reagent (Sigma). For qRT-PCR, 1–2 µg of RNA was converted to cDNA using reverse transcriptase (Applied Biosystems).

## Western blot analysis
Proteins or cell lysates were transferred to the polyvinylidene fluoride(PVDF) membrane. Blots were blocked with 2% BSA in 1X Tris-buffered saline + 0.1% Tween-20 for 1–2 h at room temperature before incubation with primary antibodies at 4 °C. The blots were incubated with the appropriate HRP-conjugated secondary antibodies and developed. To ensure equal protein loading, the same blot was either cut at the appropriate molecular weight or stripped with stripping buffer (50 mM Tris-HCl, pH 7, 2% SDS, β-mercaptoethanol) and probed with loading control antibodies. All antibodies used are listed in Supplementary Data 1.

## Generation of Rv2067c strains in *Mtb*
To generate a knockout of Rv2067c by homologous recombination, 3'(−953 to 53 bp) and the 5' regions (+1180 to + 2209 bp) of Rv2067c were amplified from the *Mtb* H37Rv genomic DNA with specific primers. 5' flanking region was cloned into SpeI and SwaI digested

pML523, followed by cloning of the 3' region into PacI and NsiI digested pML523 vector. The complete 4628 bp DNA fragment was PCR amplified and sub-cloned in pRSF-Duet digested with EcoRV digestion. This construct was transformed into wildtype *Mtb* H37Rv (Wt*Mtb*). An internal fragment of the Rv2067c ORF was replaced by a LoxP-GFP:hyg-LoxP cassette. Genomic DNA was isolated from colonies. Knockout was confirmed by PCR and western blotting with Rv2067c antibody. Rv2067c antibody was raised in rabbit and antibody specificity was checked by western analysis (Supplementary Fig. 3e). To unmark the deletion mutant, the pCreSacB plasmid was transformed into deletion mutant. Unmarking of the transformants was confirmed by PCR with hygromycin-specific primers. The unmarked strain is referred to as ΔRv2067c or knockout in the manuscript. Complemented strains were generated by transforming pST-2KRv2067c and pST-2KRv2067cRxR in ΔRv2067c background. These strains are referred to as ΔRv2067c:comp or complement and ΔRv2067c:RxR, respectively.

Overexpression strain of Rv2067c in Wt*Mtb* H37Rv was generated using inducible pST-KT vector[88]. The gene was cloned between NdeI and HindIII restriction sites. Overexpression of Rv2067c in transformants was confirmed with a concentration gradient of anhydrotetracycline (aTc) followed by immunoblotting with Rv2067c antibody. *Mtb* Rho was kept as a loading control (Supplementary Fig. 2f). The overexpression strain is referred to as Rv2067c:OE in the manuscript. Primers used are listed in Supplementary Data 2.

## Methyltransferase assay

In vitro methyltransferase assays were carried out by incubating 500 ng of methyltransferases Rv2067c or DOT1L with 1–2 μg of recombinant mammalian histones or MtHU and 80 μCi ³H-SAM (Perkin Elmer) in a buffer containing 50 mM Tris-HCl (pH 8.0), 5% glycerol, 5 mM MgCl$_2$, 1 mM dithiothreitol (DTT) and 50 mM NaCl at 37 °C for 30 min. Buffer composition for MTase assays with histone octamer and NCPs as substrates was adapted from Min et al.[8]. The reactions were stopped by adding 20% trichloroacetic acid followed by incubation on ice for 1 h. The mixture was spotted on cellulose nitrate filters (Sartorius) using a vacuum manifold and washed with 20% trichloroacetic acid, water and 95% ethanol, dried at RT, soaked in 3 ml of scintillation fluid and counts (CPMs) were measured in a scintillation counter (Perkin Elmer). For autoradiography, the reaction mixture was resolved on a 15% SDS–polyacrylamide gel electrophoresis (PAGE), electro-blotted on polyvinylidene fluoride (PVDF) membrane (GE Healthcare) and incubated for 1 h in 1% sodium salicylate solution (Sigma). The membrane was exposed to an X-ray film at −80 °C.

Salt extraction of histones was performed from THP-1 cells as described previously[89]. Briefly, 10⁷ cells were resuspended in extraction buffer with 0.2% NP-40 (10 mM HEPES pH 7.9, 10 mM KCl, 1.5 mM MgCl$_2$, 0.34 M sucrose, 10% glycerol), and the supernatant was collected as cytoplasmic extract. The nuclei pellet was lysed in no-salt buffer (3 mM EDTA, 0.2 mM EGTA), and the supernatant was collected as nuclear extract. The chromatin pellet was resuspended and sequentially dialyzed to reduce NaCl concentration to 200 mM. Extracted histones were sequentially dialyzed to reduce NaCl concentration to 200 mM. 20 μl and 10 μl of extracted histones were used as substrate for MTase using SAM as methyl donor. MTase assays with unmodified peptides were carried out using 500 ng of peptide (AnaSpec) as substrate and Rv2067c as MTase. The reactions were incubated at 37 °C for 30 min, spotted on nitrocellulose membrane and immunoblotted with H3K79me3 antibody.

## Immunofluorescence

For immunofluorescence, 0.2×10⁶ macrophages were seeded and infected with Rv2067c:OE, ΔRv2067c, *M.smeg*:FLAG and *M.smeg*:Rv2067c-FLAG. Cells were washed with PBS 24 h post infection

(h.p.i.), fixed using 4% paraformaldehyde, permeabilized with 0.1% Triton X-100 and blocked with 2% BSA. FLAG or Rv2067c antibodies were used as primary antibodies and Alexa Fluor conjugated antibodies were used as secondary antibodies. Nuclei were stained with diamidino-2-phenylindole dye (DAPI) and the cells were examined by confocal microscopy. All antibodies used are listed in Supplementary Data 1.

## Preparation of culture filtrate

Electrocompetent *M.smeg* cells were transformed with pVV16:Rv2067c-FLAG plasmid. *M.smeg* expressing Rv2067c-FLAG is referred to as *M.smeg*:Rv2067c-FLAG. 1% primary culture was inoculated in 100 ml of modified Sauton's media, and at OD$_{600}$ of 0.6–0.8, cells were harvested by centrifugation at 5000×*g* for 30 min. The supernatant was passed through a 0.45 μm filter to remove cells and concentrated using a 30 kDa cut-off centricon (Millipore). Cell pellet was resuspended in PBS and cell lysate was prepared. Secretion of Rv2067c from Wt*Mtb* strain was detected by concentrating 300 ml of culture filtrate followed by trichloroacetic acid precipitation.

## Inhibition of DOT1L

The protocol for MTase assay with DOT1L inhibitor was adapted from Daigle et al.[47]. Briefly, for in vitro assay, inhibitor EPZ004777 at a concentration up to 50 nM was incubated for 15 min with 500 ng of Rv2067c in MTase assay buffer. Post incubation, 1 μl of tritiated SAM and 2 μg of substrate were added to each of the reactions. Reactions were incubated for an additional 30 min at 37 °C and radioactive counts (CPMs) were measured in a scintillation counter. A positive control of 2 μg of reconstituted NCPs as substrate, recombinant DOT1L as MTase along with the inhibitor was kept. For assessment of EPZ004777 on Rv2067c upon infection, 3×10⁶ macrophages were seeded in a 6-well plate and exposed to varying concentrations of EPZ004777. Then, 12 h after exposure to the inhibitor, cells were infected with *M.smeg*:FLAG and *M.smeg*:Rv2067c-FLAG. Cell lysates were probed with H3K79me3 antibody 24 h.p.i. Histone H3 served as a loading control.

For RNA interference, shRNA construct targeting DOT1L was transfected into HEK293T cells and puromycin selection was applied for 24 h. RNA was isolated, followed by qRT-PCR to confirm DOT1L inhibition. DOT1L silenced HEK cells were transfected with pcDNA:Rv2067c puromycin construct and maintained under puromycin selection for another 24 h. Cell lysates were prepared and immunoblotted with H3K79me3 antibody.

## Identification of genomic loci with H3K79me3 mark performed by Rv2067c

To identify H3K79me mark specifically added by Rv2067c, HEK293T cells were co-transfected with DOT1L siRNA and pcDNA:Rv2067c-SFB using Lipofectamine 2000 (HEK-DOT1L-KD:Rv2067c). DOT1L siRNA (AM16708) and scrambled siRNA were procured from Thermo Fisher Scientific. HEK293T cells co-transfected with scrambled siRNA and pcDNA-FLAG were kept as control (HEK-scr:pcDNA). Knockdown of DOT1L in HEK cells by siRNA was confirmed by western analysis (Supplementary Fig. 14a).

The transfected cells were subjected to ChIP with H3K79me3 antibody as described by Yaseen et al.[55] with minor modifications. The enriched DNA was end-repaired using DNA polymerase I, Klenow fragment (NEB) and adapter ligated with annealed adapters (Nimblegen Systems). Next, the enriched DNA was amplified by PCR using LK102 adapter as primer and resolved on 1.5% agarose gel. The amplified product was excised, eluted and cloned in pBSK vector. After transformation, individual colonies were screened using BamHI and HindIII and positive clones were sequenced. The sequence obtained was aligned using Blat on UCSC genome browser and coordinates with enriched H3K79me3 mark were identified.

## RNA sequencing

THP-1 macrophages were infected with *M.smeg*-FLAG and *M.smeg*:Rv2067c-FLAG. 4 and 24 h.p.i, macrophages were washed with PBS and RNA was isolated using RNeasy kit (Qiagen). For transcriptomic analysis, sequencing libraries were prepared using NEB-Next® UltraTM II RNA Library Prep Kit for Illumina® following the manufacturer's instructions. Briefly, 800 ng of total RNA was used as input for poly(A) mRNA enrichment, followed by fragmentation and reverse transcription to generate cDNA. Hairpin adapter was ligated to fragmented double-strand cDNA and USER enzyme was used to cleave the hairpin structure. Ampure beads were used to purify adapter-ligated fragments and the purified product was amplified using Illumina Multiplex Adapter primers to generate a sequencing library with barcodes for each sample. RNA sequence data was generated using Illumina HiSeq. RNAseq was carried out by Clevergene, India.

## Mass spectrometry

MTase assay was carried out with recombinant histone H3 as a substrate and Rv2067c as MTase. The reaction was incubated overnight at 37 °C. The sample was resolved on 12% SDS-PAGE, gel slice was excised and subjected to MS/MS analysis along with recombinant H3 as a control. For detection of Rv2067c in the culture filtrate, concentrated culture filtrate was resolved on 12% SDS-PAGE for 15 min. 1cm of gel piece was excised and analyzed by mass spectrometry at Taplin Mass Spectrometric Facility, Harvard, USA (Orbitrap mass spectrometers, Thermo Scientific).

## Affinity pulldown

Rv2067c with N-terminal SFB-tag (S-protein, FLAG, streptavidin-binding peptide) was cloned in pcDNA3.1(pcDNA:Rv2067cSFB) and transfected into HEK293T cells using Lipofectamine 2000 (Invitrogen). A control of HEK293T cells transfected with pcDNA SFB was kept. After 24 h, cells were lysed in a buffer containing 20 mM Tris-HCl, pH 8.0, 100 mM NaCl, 1 mM EDTA, 0.5% NP-40 and protease inhibitors. Lysates were incubated with pre-equilibrated streptavidin beads (GE Healthcare) for 4 h; the beads were washed three times with the same buffer, loaded on a 12% SDS-PAGE after resuspending in 6X SDS sample loading buffer and western blotted. The blot was probed with H3 and FLAG antibodies. HEK cell line was acquired from ATCC (Manassas, VA).

For reverse immunoprecipitation (IP), the same protocol for transfection and cell lysis was followed. Cell lysates were incubated with H3 antibody coated on protein A/G beads for 4 h. IgG was used as antibody control. The blot was probed with H3 and FLAG antibodies.

Affinity pulldown using FLAG antibody on nuclear and cytosolic fractions was carried out using the IP protocol described above. using a modified protocol from Loyola et al.[76].

## Animal experiments

Six to eight weeks old male and female BALB/cJ mice were infected by aerosol with 200 bacilli per mouse with Wt*Mtb*, ΔRv2067c and ΔRv2067c:comp. At indicated times p.i, mice were euthanized, lungs were harvested for bacillary load, and tissue histopathology and granuloma scoring were performed as described before [90]. Briefly, the upper right lobe of the lungs was fixed in 10% neutral-buffered formalin. Fixed tissues were prepared as 5-μm-thick sections, embedded in paraffin, stained with hematoxylin and eosin and assessed for granuloma formation and lung damage. The remaining tissue samples were homogenized, and the bacillary load was quantified by plating serial dilutions onto Middlebrook 7H11-OADC agar plates supplemented with lyophilized BBL MGIT PANTA antibiotic mixture (polymyxin B, amphotericin B, nalidixic acid, trimethoprim, and azlocillin, as supplied by BD; USA). Colonies were counted after 3–4 weeks of incubation at 37 °C. Animal studies were carried out in strict accordance with the guidelines prescribed by the Committee for the Purpose of Control and Supervision of Experiments on Animals (CPCSEA),

Government of India. All efforts were made to minimize the suffering. Experiments were carried out in a biosafety level 3 containment facility and approved by the Institutional Animal Ethical Committee (IAEC), Indian Institute of Science (IISc), Bangalore, India (Approval number: CAF/Ethics/850/2021).

For generating Rv2067c antibody, female rabbit (New Zealand) was administered 500 μg of protein emulsified with Freud's complete adjuvant (Sigma) in 1:1 ratio. Blood from the marginal ear vein was collected post 9 days of the second booster and incubated overnight at 4 °C. The supernatant was collected and antibody titer was checked. The protocol was approved by the Institutional Animal Ethical Committee (IAEC), Indian Institute of Science (IISc), Bangalore, India (Approval number: CAF/Ethics/620/2018). The animals were maintained at the animal care facility, Indian Institute of Science. The facility adheres to animal welfare standards and guidelines. A 12 h light/dark cycle, ambient temp (23–25 °C) and humidity of 50–60% is maintained in the facility.

## Statistics and reproducibility

Statistical significance for comparison between the two groups was determined by unpaired two-tailed Student's *t*-test using GraphPad Prism software (7.0, 9.0 versions, GraphPad Software, USA) and Excel. *, *P*-value < 0.05; **, *P*-value < 0.01; ***, *P*-value < 0.001. All immunoblotting and immunofluorescence data are representative of two independent experiments. Quantitation of western blots was performed using Image J (arbitrary units normalized with the expression of the H3 or β-Actin). For MTase assay, data plotted is mean ± standard deviation for 3 independent experiments. For ChIP and qRT-PCR, the data plotted is the mean ± standard deviation of two independent sets of infections (biological replicates). For qRT-PCR, two technical replicates per sample were kept.

## Reporting summary

Further information on research design is available in the Nature Portfolio Reporting Summary linked to this article.

## Data availability

Source Data for sequencing reads are available at https://www.ncbi.nlm.nih.gov/sra/PRJNA907927 (Raw reads for RNA sequencing of THP-1 infected with *Mycobacterium* expressing Rv2067c). The coordinates for Rv2067c-SAH structure have been deposited in the Protein Data Bank under the accession code 8HKR. Other PDB codes used in this study are as follows: 1NW3 (DOT1L), AF-Q8TEK3-F1 (human DOT1L full-lengthAlphaFold2 model), 6NJ9 (Ubiquitinated nucleosome and DOT1L complex), 5E1B (NTMT1, Supplementary Fig. 9), 2NXN (PrmA, Supplementary Fig. 9), 3EGV (PrmA, Supplementary Fig. 9), 3P71 (LCMT, Supplementary Fig. 9), 5DX0 (PRMT, Supplementary Fig. 9), 6H1E (KMT9, Supplementary Fig. 9), 2F69 (SET7/9, Supplementary Fig. 9), 1KX5 (Histone H3, chain A, Supplementary Fig. 12), 5DPD (PKMT1, Supplementary Fig. 17), 5DPL (PKMT2, Supplementary Fig. 17), 3BUS (RebM, Supplementary Fig. 18). Source data are provided with this paper.

## Code availability

Source software is available at Zenodo and https://github.com/Venkat-Dadi/Rotation_Scan. CODE [https://doi.org/10.5281/zenodo.8352903] (Python script for rotation scan, data generated from rotation scan and plotting scripts). Other Python plotting script files are available with this paper as Supplementary Software Files 1–3.

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

## Acknowledgements

We thank K.N. Balaji for insightful discussions and providing cell-culture facility, S.R. Gangi Setty, N.R. Sundaresan, and S.C. Raghavan for providing reagents. We thank Harsh Bansia for initial crystallographic efforts, Ramagopal Udupi for scrutinizing the Rv2067c crystal structure, Tajinder Singh for initial MATLAB version of rotation scan script, European Synchrotron Radiation Facility (ESRF), France, for X-ray data collection and D.A. Case, Rutgers University for Amber20 licence. All experiments with *Mtb* H37Rv were carried out in BSL3 facility, Centre for Infectious Disease Research (CIDR), IISc. FACS facility at CIDR, central facility of phosphor imaging, bio-imaging and real-time PCR at the Biological Science Division, Indian Institute of Science (IISc) are acknowledged. P.R.S. is supported by a post-doctoral research associateship of Jawaharlal Nehru Centre for Advanced Scientific Research (JNCASR) and V.D. by a senior research fellowship from the University Grants Commission (UGC), Government of India. This research was supported by grants to V.N. from the Science and Engineering Research Board, Department of Science and Technology (CRG/2019/000077), and Department of Biotechnology (BT/PR13522/CoE/34/27/2015), Government of India.

## Author contributions

P.R.S., V.D., S.R., and V.N.—designed the experiments and analyzed the data. P.R.S., V.D., S.R., V.N.—wrote the manuscript. P.R.S.—performed Mtb H37Rv experiments. V.D.—performed crystallization, structure determination and computational studies. V.D., S.M.K., S.U., P.R.S.—performed protein purification. P.R.S. and S.U.—performed biochemical experiments. R.S.R., P.R.S. and S.S.—performed animal experiments. S.K.—designed *M. smegmatis* experiments. S.S. and A.S.—generation of knockout strains. V.N.—funds acquisition and overall supervision of the work.

## Competing interests

The authors declare no competing interests.
