## [Peer Review File · Nature Communications]

The Mycobacterium tuberculosis methyltransferase Rv2067c manipulates host epigenetic programming to promote its own survivalReviewer #1 (Remarks to the Author):

This MS reveals a novel function of Rv2067c, and which can secrete and methylate the host H3K79.

The data are solid, and the MS logically follows.

The MS can be accepted as is.

Reviewer #2 (Remarks to the Author):

In their manuscript entitled, "Mycobacterium tuberculosis methyltransferase manoeuvres host epigenetic programming to promote its survival," Singh et al., discover a/the function of Rv20676 as a methyltransferase and determine that this methyltransferase is involved in mycobacterium infection of the host cells. This manuscript has an enormous amount of data. The authors tested multiple histones free and in nucleosomes for the ability to methylate them and found that free H3 was the substrate in vitro. They furthermore extend these findings to cells and using primarily immunoblots, find that Rvc2067 methylates H3 upon infection. Confocal microscopy, MS, and immunoblotting showed that Rv20677 was secreted into host macrophages as well as HEK293T cells. The authors determined the crystal structure of Rv2067 via protein X-ray crystallography at 2.4 Ang resolution. The authors compare the active site to DOT1 which the authors showed was downregulated by the host and therefore may compete for the same DNA. The authors further demonstrated that the methylation added by Rv2067 alters the host gene expression, specifically these were mostly related to cytokine and cell death (necrosis/apoptosis) pathways. The authors have made several strains, including a deletion strain, and have even done some mouse experiments to show the effect 2067 makes on lungs.

The structural data is of high quality and well interpreted.

This is a wonderful piece of work and certainly warrants publication. I suggest that the authors take into consideration the following corrections/comments:

-There is a huge amount of English language problems. These problems are numerous and almost all of them relate to missing articles (i.e., a/the) or problems with singular/plural. Having these fixed is critical to publication.

-Given the authors have established an enzyme assay, it would be beneficial to add some more hard numbers such as Vmax, Kcat, etc. This would better allow for comparison to other enzymes that methylate.

-line 141 there should be a space between "comp)" and ",over"

The introduction seems too short and doesn't fully introduce all that the reader needs to know. For example, more about methyltransferases, their reaction mechanisms, what is known about DOT1, cytokines and mycobacterial infection, etc.

-Ribbons diagrams could all benefit from N- and C- termini labels

-Figure 3. AlphaFold is known to make wild loops all over the place in areas of uncertainty. The authors point this out but still present the whole predicted structure. This isn't ideal. It would be better to present just the central part of the predicted protein structure.

-The paper could really benefit from a figure which presents the proposed 2D reaction mechanism with labelled amino acid structures and electron flowing arrows. Something made in chemdraw or similar. This could then be discussed in the structural sections so the reader can easily follow the reaction.

-I can see why the authors chose to compare Rv2067 to DOT1. They are quite distant. It might be good to add a comparison to the closest relative to 2067 from TB or another bacterium. This would be useful in the structural section.

-Figure S6 was a little confusing. What do the colours mean, ie green versus grey? What stage of refinement was this from?

Reviewer #3 (Remarks to the Author):

Mycobacterium tuberculosis methyltransferase manoeuvres host epigenetic programming to promote its survival, where Singh et al have shown that how Mtb employs epigenetic modification to evade from the host defense machinery. I have few comments for this study.

Figure 1(a): How are Lane 1 and 2 different?

Figure 2: The authors have remarkably shown the tri-methylation status of free H3 by Rv2067c. However, it has been conventionally studied that different methylation patterns of the same amino acid residues such as 'mono', 'di', etc., impart different functional activities. Therefore, it is worth noting to see the effect of Rv2067c on different methylation patterns. Also, authors have shown that Rv2067c methylates H3 in the nuclear and cytosolic compartments. Is there any experiment where authors have shown Nuclear Localization Signal (NLS) is present with Rv2067c or Rv2067c is accompanied by other proteins as a part of the complex and is then translocated to the nucleus of host macrophages to carry out its function?

Figure 2i. In order to completely rule out that Rv2067c only methylates free H3, the authors should have compared the levels of H3K79me3 between the control cells and the cells transfected with Rv2067c.

Figure 3: I appreciate the authors providing such a piece of detailed knowledge on the structure of Rv2067c, and comparing it with the human methyltransferase, DOT1L. I would like to see the differences and the comparison in the amino acid composition of these two proteins. Also, it is mentioned in the manuscript that the disordered structure of DOT1L is needed for protein-protein interaction. I need a clearer explanation of how the more ordered structure of Rv2067c confers different modes of action.

Figure 7: How does EPZ004777 inhibit DOT1L. It is unclear why upon inhibition of DOT1L activity with a small molecule inhibitor in uninfected macrophages; the RNA levels of DOT1L are reduced. Please provide a clear explanation.

Does Rv2067c target other sites as well (K9 and K27) as suggested by the authors.

Figure 8 and Extended figure8: Authors have previously shown that Rv2067c methylates only free H3 and not NPC-bound H3 through the biochemical and structural data, then in these figures, they have done a ChIP assay to identify the genes impacted by H3K79 methylation by Rv2067c. These two experiments are contradictory and do not support the previous hypothesis. Authors claim that DOT1L repression influences the cytokine profile of the host. However, they have performed transcriptome of macrophages infected with M.smeg expressing Rv2067c and wt Msmeg. The effects seen can either be due to the activity of Rv2067c to directly target the histones with minimum effect of DOT1L repression. Authors should have included a set where DOT1L was inhibited.

Figure 9: What about the bacillary load in other organs like spleen? It would provide crucial information regarding dissemination of bacilli.

Is Rv2067c a druggable target?

Overall, the clear mechanism of how Rv2067c impacts host machinery is missing in this study. The link between Rv2067c-mediated repression of DOT1L, and Rv2067c-mediated methylation of free H3 is not convincing; hence, more refined experiments are needed to support the hypothesis of this manuscript. Also, in initial experiments, authors have shown that Rv2067c methylates free H3 and not the NPC-bound H3. The role of free methylated H3 in infected macrophages is

ambiguously mentioned in the manuscript. The later experiments showing the transcriptomic profiles of downregulated pro-inflammatory cytokines due to repression in DOT1L also have faulty controls.

Finally, why would Mtb downregulate a host MTase targeting H3K79 and release a protein which would target the same histone site? The Mtb protein can very well target the same set of genes targeted by DOT1L. What is the differentiating factor?

Author should update the manuscript with the new recent findings where people have shown how Mtb modulates the epigenetic machinery of the host.

Reviewer #1 (Remarks to the Author):

This MS reveals a novel function of Rv2067c, and which can secrete and methylate the host H3K79.

The data are solid, and the MS logically follows.

The MS can be accepted as is.

Response: We thank the reviewer for appreciating our substantial body of work.

Reviewer #2 (Remarks to the Author):

In their manuscript entitled, “Mycobacterium tuberculosis methyltransferase manoeuvres host epigenetic programming to promote its survival,” Singh et al., discover a/the function of Rv20676 as a methyltransferase and determine that this methyltransferase is involved in mycobacterium infection of the host cells. This manuscript has an enormous amount of data. The authors tested multiple histones free and in nucleosomes for the ability to methylate them and found that free H3 was the substrate in vitro. They furthermore extend these findings to cells and using primarily immunoblots, find that Rvc2067 methylates H3 upon infection. Confocal microscopy, MS, and immunoblotting showed that Rv20677 was secreted into host macrophages as well as HEK293T cells. The authors determined the crystal structure of Rv2067 via protein X-ray crystallography at 2.4 Ang resolution. The authors compare the active site to DOT1 which the authors showed was downregulated by the host and therefore may compete for the same DNA. The authors further demonstrated that the methylation added by Rv2067 alters the host gene expression, specifically these were mostly related to cytokine and cell death (necrosis/apoptosis) pathways. The authors have made several strains, including a deletion strain, and have even done some mouse experiments to show the effect 2067 makes on lungs.

The structural data is of high quality and well interpreted.

This is a wonderful piece of work and certainly warrants publication. I suggest that the authors take into consideration the following corrections/comments:

Response: We thank the reviewer for the appreciation of the work and constructive suggestions to improve the manuscript further.

1. There is a huge amount of English language problems. These problems are numerous and almost all of them relate to missing articles (i.e., a/the) or problems with singular/plural. Having these fixed is critical to publication.

Response: We have now carefully gone through the manuscript and language related corrections made and indicated in the revised manuscript.

2. Given the authors have established an enzyme assay, it would be beneficial to add some more hard numbers such as Vmax, Kcat, etc. This would better allow for comparison to other enzymes that methylate.

Response: Reviewers' point is appreciated. However, as already pointed out by the reviewer, we have a large amount of data (biochemical, structural, cell biology). Hence, comparison of the kinetic pattern of DOT1L on NCP and Rv2067c on free H3 would constitute an interesting new study.

3. *line 141 there should be a space between “comp)” and “,over”.*

Response: Corrected.

4. *The introduction seems too short and doesn't fully introduce all that the reader needs to know. For example, more about methyltransferases, their reaction mechanisms, what is known about DOT1, cytokines and mycobacterial infection, etc.*

Response: The reviewer's point is well taken. Now we have included additional information in the introduction that is necessary and sufficient to ensure the flow of the presentation is not disrupted (Line no. 57-71).

5: *Ribbons diagrams could all benefit from N- and C- termini labels.*

Response: Now, the N- and C- termini are labelled.

6: *Figure 3. AlphaFold is known to make wild loops all over the place in areas of uncertainty. The authors point this out but still present the whole predicted structure. This isn't ideal. It would be better to present just the central part of the predicted protein structure.*

Response: The reviewer's point is well taken. However, as we are comparing the structures of two MTases that target the same substrate but under different context, presenting the full-length DOT1L model encompassing the catalytic core whose structure is known, alongside the structure of Rv2067c gives the perspective in terms of their size differences, disordered regions (in case of DOT1L) and overall architecture. Hence, we feel the present figure could be retained as it is. It may be noted that while comparing the active site in the subsequent figures and supplementary (Fig. 4 and Supplementary Fig. 4-7 and Supplementary Fig. 19), only catalytic core is presented.

7: *The paper could really benefit from a figure which presents the proposed 2D reaction mechanism with labelled amino acid structures and electron flowing arrows. Something made in chemdraw or similar. This could then be discussed in the structural sections so the reader can easily follow the reaction.*

Response: The reviewer's point is appreciated. Now the 2D reaction mechanism has been added to the supplementary information (Page# 2, Section: Residue conservation at the active site of Rv2067c and the proposed reaction mechanism; Supplementary information).

8: *I can see why the authors chose to compare Rv2067 to DOT1. They are quite distant. It might be good to add a comparison to the closest relative to 2067 from TB or another bacterium. This would be useful in the structural section.*

Response: No paralogs of Rv2067c are found in *Mtb* or other mycobacteria. Rv2067c is structurally very similar to rickettsial protein lysine methyltransferase 1 and 2 (PKMT1/2) and we have already provided a brief comparative account of these enzymes in the original manuscript (Page#2, Section: Structural homologs of Rv2067c; supplementary information).

9: *Figure S6 was a little confusing. What do the colours mean, ie green versus grey? What stage of refinement was this from?*

Response: Figure S6 (renumbered to Supplementary Fig. 15) legend has been improved as per the reviewer's suggestion. The densities correspond to the final stage of refinement. The protein density (2Fo-Fc map, 1.5 σ) was shown in gray. The density at the SAH location was shown from the difference map (Fo-Fc map, 3.0 σ ; green) generated by refining the final structure without SAH for 20 cycles using REFMAC.

Reviewer #3 (Remarks to the Author):

Mycobacterium tuberculosis methyltransferase manoeuvres host epigenetic programming to promote its survival, where Singh et al have shown that how *Mtb* employs epigenetic modification to evade from the host defense machinery. I have few comments for this study.

1. *Figure 1(a): How are Lane 1 and 2 different?*

Response : In lane 1 and 2, two different concentrations of histones were used as can be seen in the ponceau staining. Now the amount of histones used are included in the legend.

2. *Figure 2: The authors have remarkably shown the tri-methylation status of free H3 by Rv2067c. However, it has been conventionally studied that different methylation patterns of the same amino acid residues such as 'mono', 'di', etc., impart different functional activities. Therefore, it is worth noting to see the effect of Rv2067c on different methylation patterns.*

Response : Reviewer's point is appreciated and well taken. In our *in vitro* methylation assays we could see only trimethylation. Mono- and di- methylated K79, if any, were not in detectable levels when probed with mono- and di- methylation specific antibodies. If necessary, western

blots of these data can be included. Moreover, in MS analysis we detected only H3K79me3. Hence, we pursued the current line of investigation.

3. Also, authors have shown that Rv2067c methylates H3 in the nuclear and cytosolic compartments. Is there any experiment where authors have shown Nuclear Localization Signal (NLS) is present with Rv2067c or Rv2067c is accompanied by other proteins as a part of the complex and is then translocated to the nucleus of host macrophages to carry out its function?

Response : Reviewer has raised an important point. We had thought about this and had done the experiments but not included in the manuscript as we already had a lot of data. Now we have included confocal microscopy data and immunoblots to show Rv2067c has an NLS located at its C-terminal. Deletion of 30 amino acids from C-terminal of Rv2067c inhibited its translocation into the nucleus and localised only in the cytoplasm. (Supplementary Fig. 3i and j). This information is now included in the manuscript (Line no. 162-165).

4. Figure 2i. In order to completely rule out that Rv2067c only methylates free H3, the authors should have compared the levels of H3K79me3 between the control cells and the cells transfected with Rv2067c.

Response : We do not completely understand the reviewers question and also how the suggested experiment will address the issue. To explain, we have demonstrated that free H3 is the substrate for the methylation and not H3-H4 hetero-tetramer or histone octamer or nucleosomal core particle (Fig. 1 and Supplementary Fig. 1). Structural studies also provide a rationale for such substrate-context dependent methylation complementing our biochemical results. In our considered opinion, the suggested experiment is unlikely to provide any further insight into this point, than what is provided already.

5. Figure 3: I appreciate the authors providing such a piece of detailed knowledge on the structure of Rv2067c, and comparing it with the human methyltransferase, DOT1L. I would like to see the differences and the comparison in the amino acid composition of these two proteins.

Also, it is mentioned in the manuscript that the disordered structure of DOT1L is needed for protein-protein interaction. I need a clearer explanation of how the more ordered structure of Rv2067c confers different modes of action.

Response : We thank the reviewer for appreciating the work. At the sequence level, the two MTases do not show any significant similarity. Even in the domain organization they are different as already shown in Fig. 3. Regarding the second point, already cited papers describe how disordered part of the DOT1L is involved in protein-protein interaction (Ref. 36,37 in the manuscript). Moreover, as already discussed in the manuscript, the contrasting substrate

context-dependent MTase activity of the two MTases are not correlated to their ordered structure. The reviewer may note that the catalytic trough in Rv2067c and the channel in DOT1L are both ordered. The differential activity arises from the nature of the substrate binding sites which was explained clearly in the manuscript in text and figures (Page #12, Section: Rv2067c structure precludes nucleosomal H3K79 methylation; Fig. 4).

6. *Figure 7: How does EPZ004777 inhibit DOT1L.*

It is unclear why upon inhibition of DOT1L activity with a small molecule inhibitor in uninfected macrophages; the RNA levels of DOT1L are reduced. Please provide a clear explanation.

Response : EPZ004777 is a chemical derivative of SAM. Binds to DOT1L with high affinity ($K_d = 0.1 \pm 0.02$ nM) to competitively inhibit SAM binding in the manuscript (Ref. 48 in the manuscript). This point is included in the text (Line nos. 275-277). The RNA levels of DOT1L are not reduced upon inhibitor treatment in uninfected macrophages as the reviewer has stated; no changes in the RNA level seen (renumbered to Supplementary Fig.13i, in the original submitted manuscript). Instead, what we show is that DOT1L MTase activity is inhibited by the action of the inhibitor (Fig. 7d).

7. *Does Rv2067c target other sites as well (K9 and K27) as suggested by the authors.*

Response : When lysine 79 of H3 was mutated to alanine (H3A79) and used as substrate no other methylations were detected in MTase assays (Fig. 1F). The increase in K9 and K27 mark upon *Mtb* infection at DOT1L locus would imply that these changes in the epigenetic marks are likely due to downstream consequences upon Rv2067c action. We have not suggested that Rv2067c directly targets K9 or K27.

8. *Figure 8 and Extended figure8: Authors have previously shown that Rv2067c methylates only free H3 and not NPC-bound H3 through the biochemical and structural data, then in these figures, they have done a ChIP assay to identify the genes impacted by H3K79 methylation by Rv2067c. These two experiments are contradictory and do not support the previous hypothesis.*

Response : We would like to clarify to the reviewer that there is no contradiction in our work. First, by *in vitro* assays we show that Rv2067c methylates free H3 but not reconstituted NCP (Fig.1). Second, we show that while the DOT1L expression reduced (Fig. 7a and b), H3K79me3 mark increased upon transfection and *Mtb* infection due to the MTase activity of Rv2067c (Fig. 2 b and f). Thus, as already explained in the discussion section, cytosolic histone H3 methylated by Rv2067c gets incorporated into the nucleosomes to impact the downstream

events described. It may be noted that a few host MTases methylate newly synthesised H3 and H4 before incorporation into the nucleosome (Ref. 75,76 in the manuscript). What ChIP data reveals is H3K79 mark in the chromatin context after the incorporation of methylated H3 into the nucleosomes. Also, free histones whether methylated or not do not influence gene expression.

9. Authors claim that DOT1L repression influences the cytokine profile of the host. However, they have performed transcriptome of macrophages infected with M.smeg expressing Rv2067c and wt Msmeg. The effects seen can either be due to the activity of Rv2067c to directly target the histones with minimum effect of DOT1L repression. Authors should have included a set where DOT1L was inhibited.

Response: We have performed transcriptome of macrophages infected with M.smeg expressing Rv2067c and M.smeg alone. We also validated these results in *Mtb* infection. In the manuscript we show DOT1L is repressed in THP1 macrophages infected with *WtMtb* and knockout strain complemented with Rv2067c (Δ Rv2067c:comp) (Fig. 7a). Then, by performing ChIP with H3K79me3 antibody on infected macrophages, we show reduced enrichment of H3K79me3 mark on the promoter region of IL-6 and TNF- α along with the concomitant reduced expression of these cytokines (Fig. 8c and d). An earlier study has shown the reduction in cytokine level upon DOT1L inhibition (Ref. 51 in the manuscript). If the cytokine profile was influenced directly by Rv2067c activity, the H3K79me3 enrichment and cytokine expression would have been retained upon infection.

10. Figure 9: What about the bacillary load in other organs like spleen? It would provide crucial information regarding dissemination of bacilli.

Response : We thank the reviewer for this point. We have not analysed the spleen cfu for the following reason. All the studies described are confined to macrophages and lung infections. As a result investigation in mice was confined to lungs.

11. Is Rv2067c a druggable target?

Response : Very likely. Hence, we had suggested this point in the last line of discussion in the original version itself. This is certainly an important point and we would like to pursue this line of investigation in future.

12. Overall, the clear mechanism of how Rv2067c impacts host machinery is missing in this study. The link between Rv2067c-mediated repression of DOT1L, and Rv2067c-mediated methylation of free H3 is not convincing; hence, more refined experiments are needed to

support the hypothesis of this manuscript. Also, in initial experiments, authors have shown that Rv2067c methylates free H3 and not the NPC-bound H3. The role of free methylated H3 in infected macrophages is ambiguously mentioned in the manuscript. The later experiments showing the transcriptomic profiles of downregulated pro-inflammatory cytokines due to repression in DOT1L also have faulty controls.

Response : In light of reviewer 1 and 2 comments and our own long sustained efforts to understand how Rv2067c manipulates host signalling we beg to disagree with the reviewer's points. We have shown Rv2067c mediated methylation of free H3 by *in vitro* assays (Fig. 1h and Supplementary Fig. 1i), *Mtb* infection and transfection (Fig. 2h and i). We also show structure based context dependent specificity of Rv2067c for free H3 (Fig. 4 and Fig. 6). It would have been desirable to get clear suggestions to carry out more refined experiments. Further, to explain Rv2067c-mediated repression of DOT1L upon *Mtb* infection, we show by ChIP-qRT that it could be due to increased repressive methylation marks (H3K9me3 and H3K27me3) on the DOT1L gene. As responded to the reviewer 2 and in our own interest we are carrying out investigations to further understand Rv2067c action. Clearly, all these experiments are beyond the scope of present study.

13. Finally, why would Mtb downregulate a host MTase targeting H3K79 and release a protein which would target the same histone site? The Mtb protein can very well target the same set of genes targeted by DOT1L. What is the differentiating factor?

Response : The reviewer may note that if Rv2067c were to directly target same set of genes targeted by DOT1L (IL-6 and TNF- α), the consequence would be different. As already explained in response to question 9, proinflammatory cytokine profile will not be reduced. It may be noted that H3K79me3 is an activating mark and DOT1L-mediated K79 mark increases the cytokine expression. Clearly, *Mtb* has evolved a clever strategy to repress the signals from host MTase by repressing the host enzyme and adding new K79 marks on subset of genes (where DOT1L marks are not found) such as NLRC3, SESTRIN3 and TMTC1 which steers the pathogen to withstand host's arsenals. Hence, as already illustrated in the manuscript *Mtb* employs multi-pronged approach (Fig.10) to modify host machinery by intercepting the host epigenetic circuit for its survival. To the best of our knowledge, no other pathogen effector protein known exerts multiple effects in host.

14. Author should update the manuscript with the new recent findings where people have shown how Mtb modulates the epigenetic machinery of the host.

Response: Reviewer's point is noted and recent finding are included in the discussion.

Reviewer #2 (Remarks to the Author):

The authors have satisfactorily answered the reviewers comments and the manuscript can be published now.